# State-dependent representations of mixtures by the olfactory bulb

Aliya Mari Adefuin†, Sander Lindeman†, Janine Kristin Reinert†, Izumi Fukunaga*

Sensory and Behavioural Neuroscience Unit, Okinawa Institute of Science and Technology Graduate University, Okinawa, Japan

**Abstract** Sensory systems are often tasked to analyse complex signals from the environment, separating relevant from irrelevant parts. This process of decomposing signals is challenging when a mixture of signals does not equal the sum of its parts, leading to an unpredictable corruption of signal patterns. In olfaction, nonlinear summation is prevalent at various stages of sensory processing. Here, we investigate how the olfactory system deals with binary mixtures of odours under different brain states by two-photon imaging of olfactory bulb (OB) output neurons. Unlike previous studies using anaesthetised animals, we found that mixture summation is more linear in the early phase of evoked responses in awake, head-fixed mice performing an odour detection task, due to dampened responses. Despite smaller and more variable responses, decoding analyses indicated that the data from behaving mice was well discriminable. Curiously, the time course of decoding accuracy did not correlate strictly with the linearity of summation. Further, a comparison with naïve mice indicated that learning to accurately perform the mixture detection task is not accompanied by more linear mixture summation. Finally, using a simulation, we demonstrate that, while saturating sublinearity tends to degrade the discriminability, the extent of the impairment may depend on other factors, including pattern decorrelation. Altogether, our results demonstrate that the mixture representation in the primary olfactory area is state-dependent, but the analytical perception may not strictly correlate with linearity in summation.

**\*For correspondence:**
izumi.fukunaga@oist.jp

†These authors contributed equally to this work

**Competing interest:** The authors declare that no competing interests exist.

## Editor's evaluation

This study provides a strong evidence supporting that odour mixture interactions are more linear during awake animals compared to anaesthetised conditions, contrary to the previous notion that odour mixture interactions are sublinear – the conclusion obtained in anaesthetised animals. With their revisions, the authors have done new simulations to clarify how linear mixture interactions affect odour discriminability and interact with other factors (e.g. decorrelations). This new analysis provides a reasonable explanation as to when linearity helps improve discriminability and clarify the significance of sublinear interactions.

## Introduction

As animals in nature navigate through their environment in order to find food, mates, and to avoid dangers, their sensory systems are tasked to detect and recognise signals of interest despite a background of interfering signals. This figure-ground segregation is a ubiquitous task for many, if not all, sensory systems. In the visual system, for example, segmentation of spatial patterns of light allows animals to recognise objects despite some parts being obscured (*Marr, 1982*). In the auditory system, spectral combinations of sound waves are recognised and strung together over time to form a stream, allowing animals to recognise social calls from specific individuals among other noises (*Bregman,*

*1990*). In olfaction, too, animals face challenges in identifying an odour of interest in the presence of other molecules (*Laing and Francis, 1989*; *Rokni et al., 2014*).

Olfactory stimuli are first detected by a large family of olfactory receptors residing in the nasal epithelium (*Buck and Axel, 1991*). Due to a broad ligand-receptor binding (*Araneda et al., 2000*; *Del Mármol et al., 2021*; *Malnic et al., 1999*), each odour molecule may activate a number of olfactory receptor types, leading to a combinatorial representation (*Malnic et al., 1999*). As a result, when several compounds are present in a given mixture, they can activate overlapping sets of olfactory receptors, causing complex pharmacological interactions. For example, molecules may bind a common receptor, which, depending on the efficacy, can lead to antagonism (*Cruz and Lowe, 2013*; *Kurahashi et al., 1994*; *Oka et al., 2004*; *Reddy et al., 2018*; *Singh et al., 2019*) or enhancement (*Xu et al., 2020*). Recent large-scale studies demonstrate that this is a widespread phenomenon (*Inagaki et al., 2020*; *Xu et al., 2020*; *Zak et al., 2020*), which means that neural responses to mixtures often do not equal the sums of responses to the individual odours. In addition to the interactions at the periphery, there are many forms of nonlinear summation at multiple stages of olfactory processing, including the saturation of neural responses (*Firestein et al., 1993*; *Wachowiak and Cohen, 2001*) and inhibitory interactions within the olfactory bulb (OB) (*Economo et al., 2016*). Widespread suppressive interactions are also observed in downstream areas, including the piriform cortex (*Penker et al., 2020*; *Stettler and Axel, 2009*).

Nonlinear summation of signals in some brain areas may be desirable when specific combinations of signals carry special meanings (*Agmon-Snir et al., 1998*; *Jacob et al., 2008*). However, in primary sensory areas, it is sometimes considered information limiting (*Laughlin, 1989*). For olfaction, this is thought to limit the analytical ability – whether a mixture *can* be perceived in terms of the constituent qualities (*Bell et al., 1987*; *Jinks and Laing, 1999*; *Laing and Glemarec, 1992*). Nonlinear summations, or 'interactions', do not occur for all odour mixtures (*Fletcher, 2011*; *Gupta et al., 2015*; *Tabor et al., 2004*), but occur prevalently when the background and target activation patterns overlap. This happens when a mixture contains many components (*Mathis et al., 2016*), as well as when the background odours are structurally related to the target odour (*Cruz and Lowe, 2013*; *Fletcher, 2011*; *Jinks and Laing, 1999*; *Kay et al., 2003*; *Mathis et al., 2016*; *Tabor et al., 2004*). Nonlinear summation poses a difficulty because it may distort a pattern of interest brought by non-uniform addition of unpredictable background patterns, although more recent studies suggest beneficial effects of antagonism, for example, by reducing saturation-related loss of information (*Reddy et al., 2018*). Due to this difficulty, some studies have suggested that the olfactory system may not decompose mixture representations into component parts, but may solve the task by learning task-specific boundaries instead (*Mathis et al., 2016*; *Wilson and Stevenson, 2003*).

The question therefore remains: how does the mammalian olfactory system deal with nonlinear summation of responses? To investigate this, we used binary mixtures of odours to investigate mixture representations in mice performing an odour detection task. While temporal structures caused by turbulence may be used to segregate odours of interest from the background (*Ackels et al., 2021*; *Hopfield, 1991*), we tackle the case when temporal information from the environment is not available. We demonstrate, by comparing the mixture responses using various paradigms, that the property of mixture summation depends on the brain state.

## Results

Olfactory figure-ground segregation is most difficult when the target and background odours evoke overlapping activity patterns (*Rokni et al., 2014*). To study this task using binary mixtures, we first characterised how our target odour relates to other odours in our panel. The target odour was ethyl butyrate (EB) and the rest of the odours in our set comprised a range of small esters structurally similar to EB, as well as non-esters (*Figure 1A*). According to Rokni et al., the masking index, which measures the amount of overlap in response patterns between the target odour and background odours, correlates well with behavioural performances (*Rokni et al., 2014*).

To measure the masking indices from single-odour responses, we obtained activity patterns from the OB output neurons, the mitral and tufted (M/T) cells. Using a two-photon microscope, we imaged from the glomerular layer of the OB in *Tbx21*[Cre]::Ai95D mice, which express the calcium indicator GCaMP6f (*Dana et al., 2019*) in M/T cells (*Haddad et al., 2013*). To reproduce previous results, these initial imaging experiments took place in mice under anaesthesia (*Figure 1B*). We studied the degree

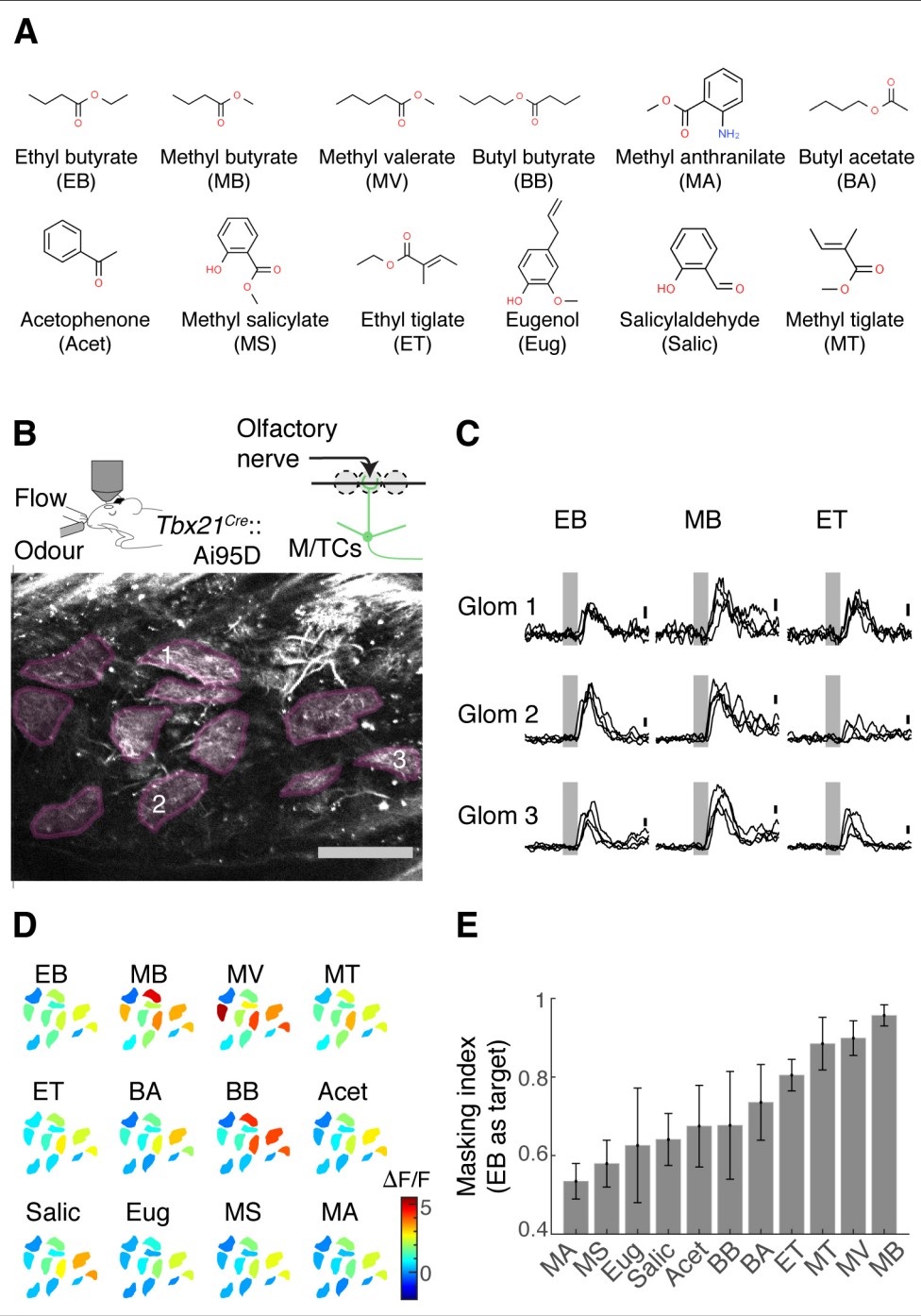

**Figure 1.** Masking indices of odours used with respect to the ethyl butyrate (EB) pattern. (**A**) Odours in the panel with abbreviations used in the rest of the article. EB was the target odour for behavioural experiments. (**B**) Two-photon imaging of GCaMP6f signals from the apical dendrites of mitral and tufted (M/T) cells in *Tbx21^Cre^::Ai95D* mice under ketamine and xylazine anaesthesia. Scale bar = 0.1 mm. (**C**) Example GCaMP6f transients expressed as a change in fluorescence (ΔF/F). Scale bar = 1 ΔF/F. Grey, odour presentation (0.5 s). (**D**) Example evoked responses. Manually delineated regions of interest (ROIs) are shown with fluorescence changes evoked by odours, indicated with the corresponding colour map. The amplitude indicated is the average change during 1 s from the final valve opening. (**E**) Masking indices for all odours in the panel, with EB as the target. N = 4 fields of view, four mice. Mean and SEM of three trials or more shown.

of overlap between EB and other odour responses by presenting single odours in a randomised order. An analysis of glomerular activity patterns revealed that methyl butyrate (MB) responses overlap the most with the EB response patterns, followed by closely related esters (*Figure 1C–E*).

Previous studies from anaesthetised animals showed that neural responses to odour mixtures exhibit widespread nonlinear summation (*Inagaki et al., 2020*; Oka et al.; *Reddy et al., 2018*; *Singh et al., 2019*; *Xu et al., 2020*). In particular, suppressive interactions become dominant among large responses due to saturation (*Mathis et al., 2016*). This pattern is observed at many stages of olfactory processing, including in the OB (*Economo et al., 2016*; *Fletcher, 2011*) and anterior piriform cortex (*Penker et al., 2020*), but it depends on the complexity of mixtures, as well as odorant choices (*Fletcher, 2011*; *Gupta et al., 2015*; *Rokni and Murthy, 2014*; *Tabor et al., 2004*). We therefore characterised the property of binary mixture summation using our odour set.

We first confirmed that our olfactometer is capable of presenting stable concentrations of odours for mixtures. We used a photoionisation detector to ensure that, when two odours are mixed, ionisation levels sum linearly (*Figure 2A–C*). Then, to assess how the OB output represents binary mixtures, we imaged the individual somata of M/T cells in *Tbx21^Cre^::Ai95D* mice (*Figure 2D*). A typical session consisted of about 40 trials to minimise time-dependent effects. Single odours and their binary combinations were presented in a semi-random order. Since EB, MB, and the mixture of these two odours are of particular importance in this study, these three trial types appeared every 10 trials (*Figure 2E*; see Materials and methods). To assess how the mixture of EB and MB is represented, the amplitudes of the mixture responses were compared against those of linear sums of single-odour responses (*Figure 2F and G*). As reported before, a large proportion of M/T cells exhibited nonlinear summation (39.0% of M/T cells showed a deviation from linearity greater than 2; 71/182 regions of interest [ROIs], seven fields of view, four mice; see Materials and methods) with a large fraction showing sublinear summation (35.7%, 65/182 ROIs).

Given that nonlinear summation is so widespread in the OB output, how well can mice analyse binary mixtures to accurately detect the presence of the target odour at the behavioural level? To assess this, we used a Go/No-Go paradigm for head-fixed mice. The rewarded stimulus (S+ odour) contained EB, either as a single odour or as a component of binary mixtures (*Figure 3A*). To train mice on this task efficiently, after habituation, the head-fixed mice were first trained to discriminate EB against other single odours (*Figure 3A*). The mice learned to perform this task well, reaching an accuracy of 80% within 200 trials on average (*Figure 3B and C*; number of trials to reach 80% accuracy = 194.3 ± 21.9; n = 7 mice). Subsequently, these mice were trained to detect the presence of EB in binary mixtures (*Figure 3B and C*). The mice performed the mixture task at a high accuracy from the beginning (90.2% ± 1.3% of trials with correct responses in the first session; n = 7 mice). However, the mistakes they made were odour-specific, in that they tended to lick more indiscriminately on MB-containing trials regardless of the presence of EB (*Figure 3D and E*; mean lick preference index for MB trials = 0.74 ± 0.04 vs. 0.93 for five other background odours, p = 0.0035, one-way ANOVA; n = 7 mice). However, with training, the performance on MB trials, too, became accurate, demonstrating that mice can learn to accurately detect the target odour in binary mixtures even when the background odour is similar.

To understand why mice were able to acquire the binary mixture task so easily, we analysed how M/T cells represent binary olfactory mixtures while the trained mice accurately performed the task. GCaMP6f fluorescence was measured from M/T cell somata. Data from the behaving mice was compared to the case when the same mice were later anaesthetised with ketamine and xylazine (*Figure 4A–D*). This revealed that the M/T cells in behaving mice tended to sum mixture responses more linearly than when the mice were anaesthetised (*Figure 4D*). To quantify this data, we expressed a deviation from the linear sum as a fraction of the predicted, linear sum of component responses ('fractional deviation'; *Figure 4E*). Data from awake, behaving mice showed a clear shift in the distribution of this index compared to the anaesthetised case. In the anaesthetised case, a steep slope in the cumulative histogram occurs at a negative value (median fractional deviation = –0.39), indicating that the majority of the OB output neurons exhibit sublinear summation. On the other hand, M/T cells from behaving mice showed a broader distribution centred around 0 (median fractional deviation = –0.12; $P = 3.83 \times 10^{-10}$, two-sample Kolmogorov–Smirnov test comparing the behaving vs. anaesthetised data; n = 202 ROIs and 103 ROIs, respectively). The broader distribution for the awake, behaving condition likely reflects a greater variability in this condition. Consistent with this, adding

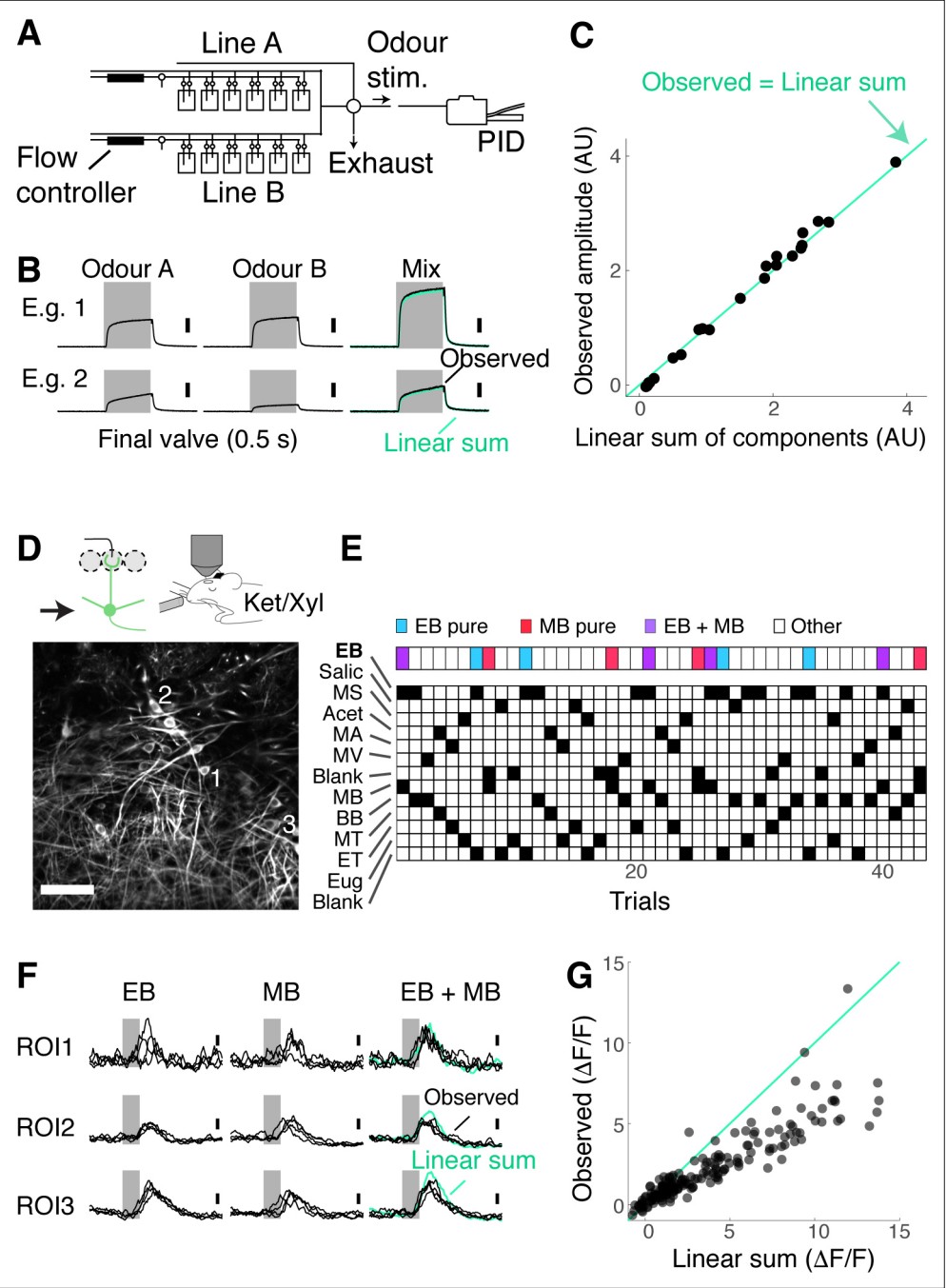

**Figure 2.** Mixture suppression dominates under anaesthesia. (**A–C**) Validation of linear mixing by the olfactometer. (**A**) Binary mixtures are generated by mixing odorised air from two streams each equipped with a mass flow controller and presented as a stimulus when a five-way valve ('final valve') is actuated. To present single odours, odorised air from one line was mixed with air that passes through a blank canister in the other line. A photoionisation detector (PID) was used for calibration. (**B**) Example PID measurements for single odours ('Odour A' and 'Odour B'), and their mixtures. Linear sum of the components (green trace) is shown superimposed with the observed PID signal (black). (**C**) Observed amplitudes for mixtures vs. linear sum of component amplitudes. Light blue line, the observed mixture amplitude equals the linear sum of components. (**D–G**) Investigation of mixture summation by mitral and tufted (M/T) cells under anaesthesia. (**D**) Top: schematic of two-photon GCaMP6f imaging from somata of M/T cells in naïve *Tbx21^{Cre}*::Ai95D mice under ketamine/xylazine anaesthesia. Bottom: example field of view. Scale bar = 50 μm. (**E**) Example session structure. Single ethyl butyrate (EB) and methyl butyrate (MB) trials, as well as EB + MB mixture trials, are indicated with colour codes. (**F**) Transients from three example regions

*Figure 2 continued on next page*

*Figure 2 continued*

of interest (ROIs) as highlighted in (**D**). Linear sum (green trace) was constructed by linearly summing the averages of single EB and MB responses. Grey bar represents the time of odour presentation (0.5 s). (**G**) Scatter plot of observed mixture response amplitude against linear sum of components. N = 183 ROIs, seven fields of view, four mice.

---

a Gaussian noise to the data from anaesthetised mice made the distribution broader, but without affecting the median (*Figure 4F*). Using this index to compare the two conditions, we found that the difference was most striking in the first second after the odour onset ('early phase'; *Figure 4G*). Further, we observed that the responses in the somata are more linear in the early phase compared to the apical dendrites (*Figure 4—figure supplement 1*). These observations together may indicate that dampened responses in the early phase of somatic responses may underlie the reduced mixture suppression, although some contribution may come from more decorrelated responses at the somata (*Figure 4—figure supplement 1*). Overall, the results indicate that the property of mixture summation is more linear in the awake, behaving mice, and that linearisation is not imprinted permanently as a result of learning.

While the above result demonstrates that mixtures sum more linearly in awake, behaving mice, M/T cell responses are also significantly smaller and more variable in this condition. This is quantified in an analysis of trial-by-trial response similarity, which measures the Pearson correlation coefficient between population responses from individual trials (*Figure 5A–C*). This showed that, in the anaesthetised mice, the M/T cell response patterns are highly correlated, both within class (comparison between trials with EB) and across classes (*Figure 5A and B*). In the awake, behaving mice, as expected, responses were less consistent, indicated by the lower within-class correlation (*Figure 5A and B*). However, because across-class patterns were substantially less correlated (*Figure 5A and B*), S+ responses may be more discriminable in the behaving mice.

The ability to reliably encode the presence of EB was first quantified based on within-class vs. across-class correlation by counting the proportion of trials where within-class correlation was higher than across-class (*Bridgeford et al., 2021*). This indicated that M/T cells from anaesthetised mice showed better discriminability initially, but later, M/T cells in the behaving mice performed as well as the anaesthetised mice (*Figure 5C*). Note that a similar result is obtained when trials are sorted by the presence of MB (*Figure 5—figure supplement 1*). While useful, the Pearson correlation coefficient takes into account all ROIs, irrespective of how informative they are in discriminating patterns, and may not reflect the true ability of a population of neurons to encode odours. To address this, we trained support vector machines (SVMs) to discriminate responses evoked by odours containing EB (S+) vs. odours without EB (S-). We used 80% of randomly selected trials from each session for training, and the remaining 20% of the trials to test the performance (*Figure 5D*). This time, M/T cells from the two conditions performed similarly for the first 1 s, performing above chance soon after the odour onset (earliest time where accuracy is significantly above 0.5 = 0.67 s for the behaving case, and 1.12 s for anaesthetised data; *t*-test at a significance level of 0.05; n = 13 fields of view, six mice for behaving case; 8 fields of view, four mice for the anaesthetised case; *Figure 5D*). Further, the data from behaving mice outperformed the anaesthetised case in the later phase (median accuracy for 2–3 s after odour onset = 0.58 vs. 0.70 for anaesthetised vs. behaving mice, respectively; p = 0.027, Mann–Whitney *U* test for equal medians). We also wished to assess how mixture responses relate to single-odour patterns. To do so, we trained SVMs using single-odour responses (EB vs. other single odours) and tested the performance on mixture responses (*Figure 5E*). Again, the data from behaving mice outperformed the data from anaesthetised mice in the late phase, although the difference was not statistically significant (median accuracy for 2–3 s after odour onset = 0.55 vs. 0.64 for anaesthetised vs. behaving mice, respectively; p = 0.065, Mann–Whitney *U* test for equal medians). This may suggest that, over time, mixture responses come to resemble the target component odour pattern more in the awake, behaving mice than in the anaesthetised mice. Curiously, the time course of decoder accuracy did not strictly depend on the linearity of mixture summation (*Figure 6—figure supplement 2*). Overall, the result suggests that, despite the substantial differences in the amplitudes and variability of responses, M/T cells in behaving mice are able to encode the presence of the target odour well.

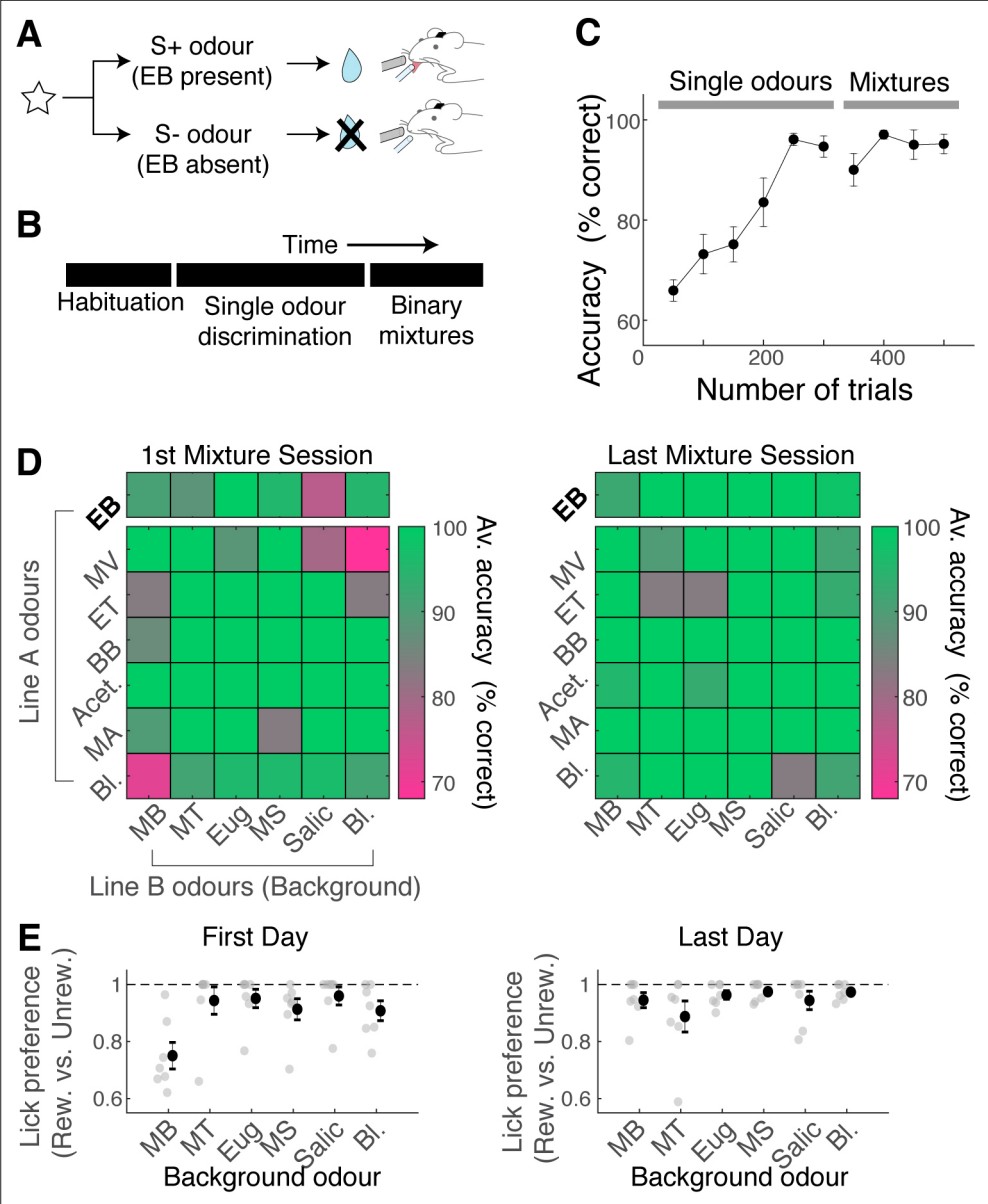

**Figure 3.** Mice can learn to accurately analyse difficult binary mixtures. (**A**) Behavioural paradigm: Go/No-Go task with head-fixed mice. Ethyl butyrate (EB)-containing olfactory stimulus was the rewarded stimulus. (**B**) After habituation, mice learned to discriminate EB against other single odours in the panel. Once proficient, mice learned to detect the presence of EB in binary mixtures. 30% of stimuli in the mixture stage were single odours. (**C**) Behavioural performance for all mice (n = 7 mice). Mean and SEM shown. (**D**) Odour-specific accuracy shown for the first (left) and last day of mixture training (right). Green shades indicate high accuracy. Top row corresponds to rewarded trials, and bottom six rows correspond to unrewarded trials. Average accuracy from all animals shown (n = 7 mice). Bl. = blank. (**E**) Lick preference index measures licks that occur preferentially on rewarded trials for a given background odour. A lick preference value of 1 occurs when all anticipatory licks were observed in rewarded trials only. Grey points, data from individual animals (n = 7 mice); mean and SEM shown in black.

To what extent is the mixture representation in the OB affected by learning? Previous studies suggest that prior exposures and familiarity to odours affect the ability to analyse odour mixtures (*Grabska-Barwińska et al., 2017*; *Poupon et al., 2018*). To address this, we assessed how EB and MB mixtures are represented in naïve but awake, head-fixed mice. Since different levels of motivation, with accompanying changes in the sniff patterns, affect the olfactory system (*Carey and Wachowiak, 2011*; *Jordan et al., 2018*), we made one group naïve mice engaged and a second group of mice

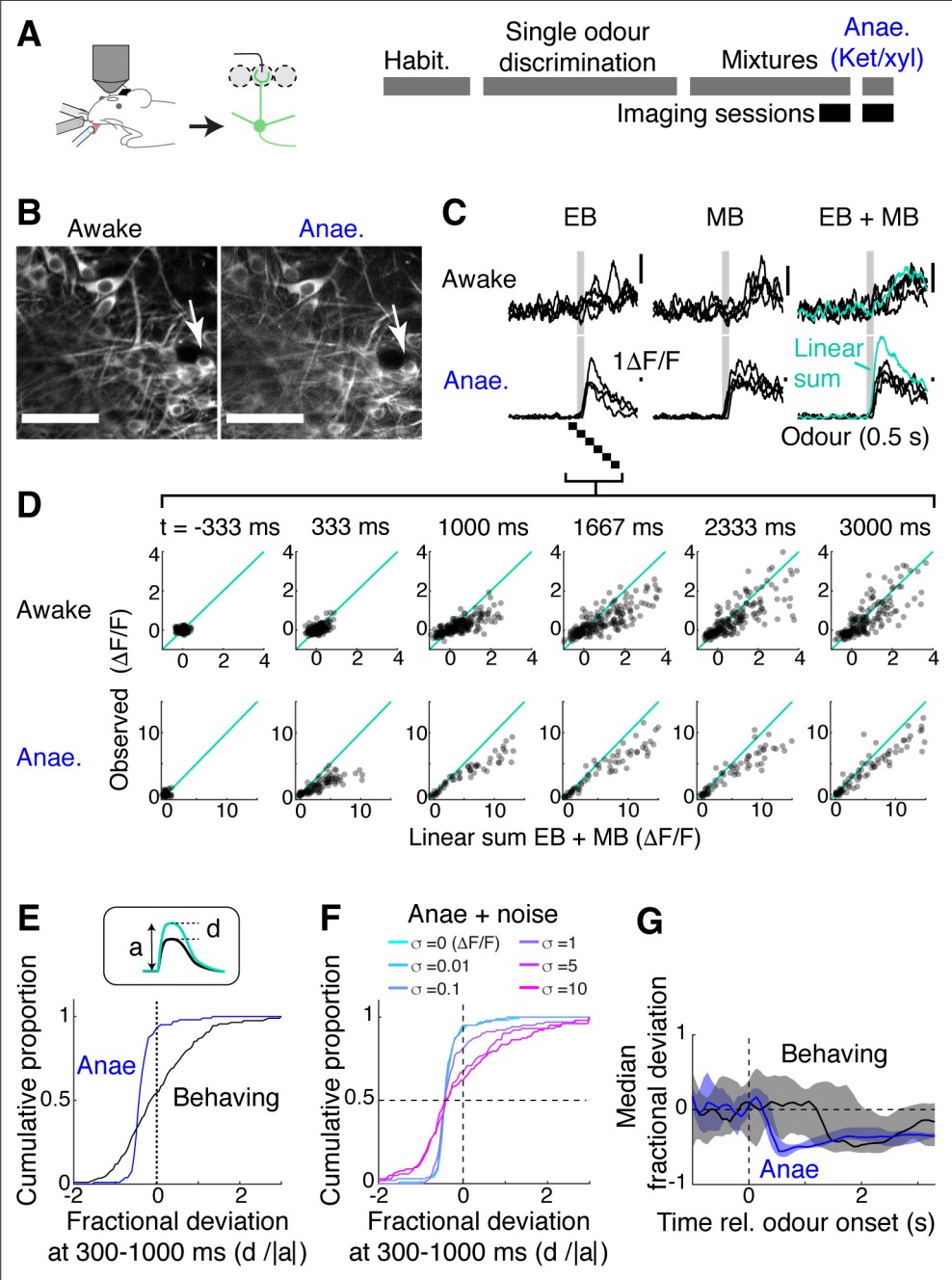

**Figure 4.** Mixture summation is more linear in awake, behaving mice. (**A**) Schematic of experimental setup. Imaging of GCaMP6f signals from the somata in *Tbx21^Cre*::Ai95D mice performing the mixture detection task. On the last day, imaging took place under ketamine/xylazine anaesthesia (Anae). (**B**) Example field of view showing the same neuron from two imaging sessions. (**C**) Relative fluorescence change evoked by ethyl butyrate (EB), methyl butyrate (MB), and their mixture for the two conditions for the neuron indicated by arrow in (**B**). Grey bar, odour presentation (0.5 s). (**D**) Scatter plot of observed EB + MB mixture response amplitude against linear sum of component odour responses (average of 20 frames). Indicated time is relative to odour onset. (**E**) Cumulative histograms of fractional deviation from linear sum for data from anaesthetised mice (blue) and trained mice performing mixture detection task (black). N = 202 regions of interest (ROIs) from 13 fields of view, six mice for behaving case; 103 ROIs from eight imaging sessions, four mice for the anaesthetised case. (**F**) Effect of Gaussian noise on the fractional deviation distribution. Colours correspond to data with different amount of noise added. (**G**) Time course of median fractional deviation for the two datasets. Median deviation was obtained from each

*Figure 4 continued on next page*

*Figure 4 continued*
field of view; thick line shows the median for all fields of view, along with the 25th and 75th percentiles (shaded areas). See also *Figure 4—figure supplement 1*.

The online version of this article includes the following figure supplement(s) for figure 4:

**Figure supplement 1.** Subcellular dependence on mixture summation properties.

disengaged by changing the reward contingency (*Figure 6A*). That is, we presented the same sets of odour stimuli, but instead of associating specific odours with reward, the water reward was either delivered after odour onset but on randomly selected trials to engaged the mice, or about 15 s before the odour onset on all trials in order to disengage the mice (*Figure 6A*; n = 17 sessions, six mice, and 14 sessions, four mice, respectively). The level of engagement was confirmed by the speed of inhalations during odour presentations (*Figure 6B*) and the number of anticipatory licks generated (*Figure 6—figure supplement 1*).

In all awake conditions, we found the EB and MB responses to sum equally linearly (*Figure 6—figure supplement 1*), indicating that the linearity of summation does not depend on the behavioural states or learning. However, the performance of SVMs on these datasets showed a state-dependence. While the general ability to discriminate S+ vs. S- odours was comparable across the states, the ability to analyse mixtures in terms of the constituents – SVMs trained on single odours and tested on mixture patterns – was particularly poor for the disengaged mice (*Figure 6C–F*). Since the linearity of summation was comparable across the three groups (*Figure 6—figure supplement 1*), this, again, suggests that the ability to correctly decode the presence of an odour does not strictly correlate with the linearity of mixture summation. In all cases, the decoder performed most accurately in the late phase. We hypothesised that this extra time may reflect an involvement of recurrent interactions with the piriform cortex, which is thought to store learned representations and feed back to the OB (*Chapuis and Wilson, 2011*; *Grabska-Barwińska et al., 2017*). Our preliminary result from muscimol infusion in the ipsilateral piriform cortex, however, suggests that this may not be the case (*Figure 6—figure supplement 3*).

Under what circumstance does discriminability of odour mixtures decouple from linearity in mixture summation? To study the effect of saturating sublinearity in isolation, we made a simulation as follows. Linear sums of responses were constructed by adding component responses obtained from the imaging sessions. In one test, these sums, with added noise, were passed through SVMs that had been trained with single odour responses. In another test, the linearly summed responses with noise were transformed with a normalising function (*Mathis et al., 2016*; *Penker et al., 2020*), which adds sublinearity in an amplitude-dependent manner (*Figure 7B*; see Materials and methods). Then, these signals were passed through the same SVMs to assess how discriminable the activity patterns were. When the data from the anaesthetised mice was used, normalising sublinearity was particularly detrimental around 2 s after the odour onset (*Figure 7C*), qualitatively reproducing the transient decrease in the accuracy seen with the observed mixture responses (*Figure 5E*). In contrast, with the data from the behaving mice, normalising sublinearity did not have as significant an effect on the mixture discriminability, even in the late stage when sublinear summation becomes more widespread (*Figure 7D and E*). Thus, the functional consequence of nonlinear summation may depend on specific circumstances, for example, on how separable, or decorrelated, the activity patterns are. Overall, our result indicates that mixture responses in the OB are highly state-dependent and evolve over time (*Figure 6F*).

## Discussion

Segmentation and extraction of relevant information from mixtures of signals are important and perpetual tasks for sensory systems. Nonlinear summation of signals is generally thought to pose difficulty in demixing component signals. Here, we demonstrate that the signal summation in the primary olfactory area of the mouse is significantly more linear in awake mice. This is notable because many previous studies reported that suppressive mixture interactions are widespread in the central nervous system, based on anaesthetised preparations (*Bell et al., 1987*; *Cruz and Lowe, 2013*; *Penker et al., 2020*; *Stettler and Axel, 2009*).

However, our analysis also indicates that the ability to discriminate binary mixture responses does not correlate strictly with linearity in summation. First, the behavioural improvement was not

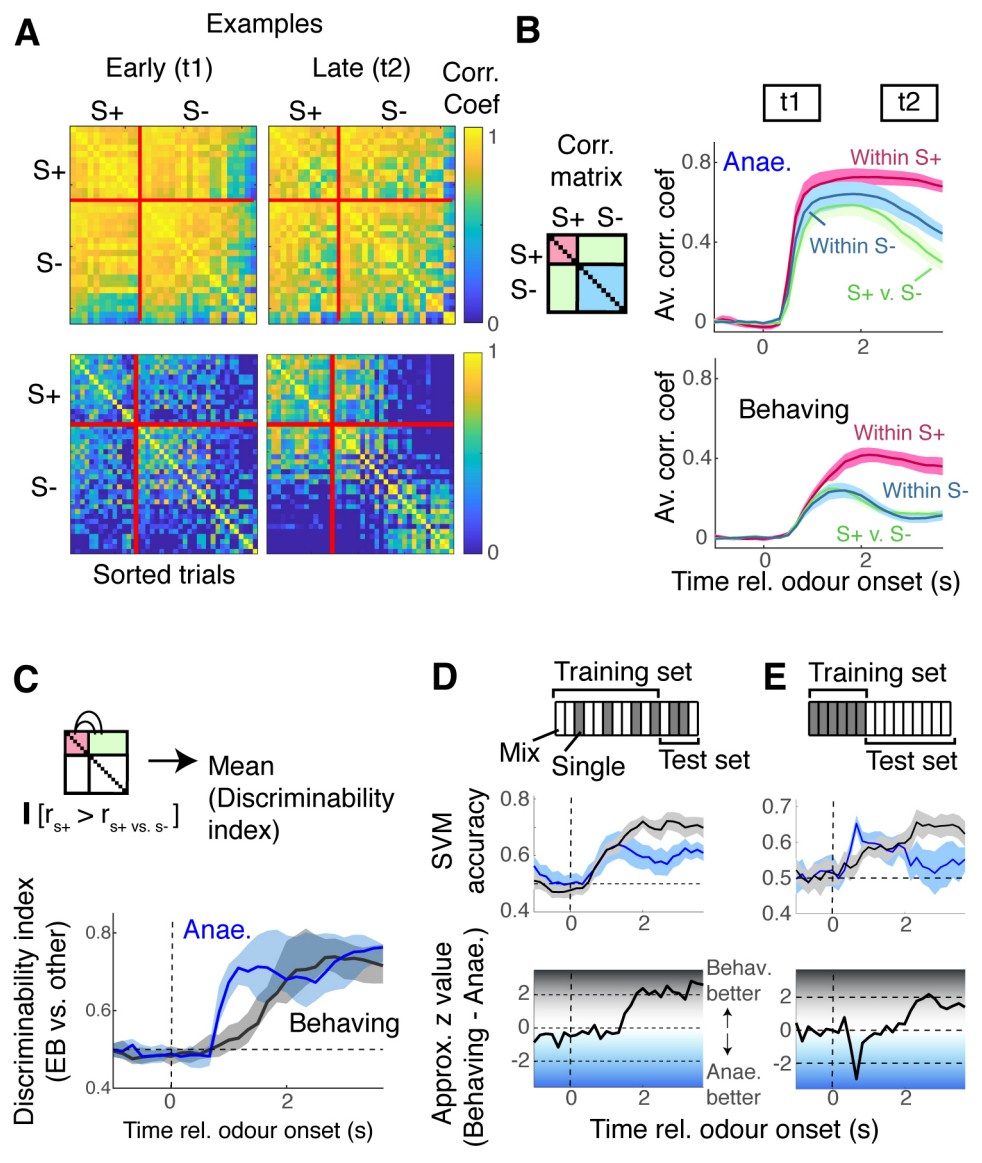

**Figure 5.** Encoding the presence of the target odour is better in behaving mice. (**A**) Correlation matrices for data from anaesthetised mice (top) and awake, behaving mice (bottom), similarity of evoked responses between individual trials, for two time points, t1 and t2, as indicated in panel (**B**). S+ trials were trials with ethyl butyrate (EB) either as single odour or in binary mixtures. (**B**) Average correlation coefficient within S+ trials (magenta), within S- trials (blue), or across trials (green) for mitral and tufted cells (M/TC) responses from anaesthetised mice (top panel) and awake, behaving mice (bottom panel). (**C**) Discriminability index, which measures the proportion of trials in which the within-S+ correlation was higher than the across class, was compared at each time point for data from anaesthetised mice (blue trace) and awake, behaving mice (black). See also *Figure 5—figure supplement 1*. (**D**) Support vector machine (SVM) for discriminating EB vs. non-EB responses, trained on M/TC somatic responses from randomly selected 80% of trials, and tested on 20% of trials for anaesthetised (blue) and behaving (black) conditions. Both training data and test data contained random selection of single-odour responses (grey rectangles in scheme) and mixture responses (hollow rectangles). (**E**) SVM trained with single-odour responses, tested on mixture responses.

The online version of this article includes the following figure supplement(s) for figure 5:

**Figure supplement 1.** Discriminability of methyl butyrate (MB) responses.

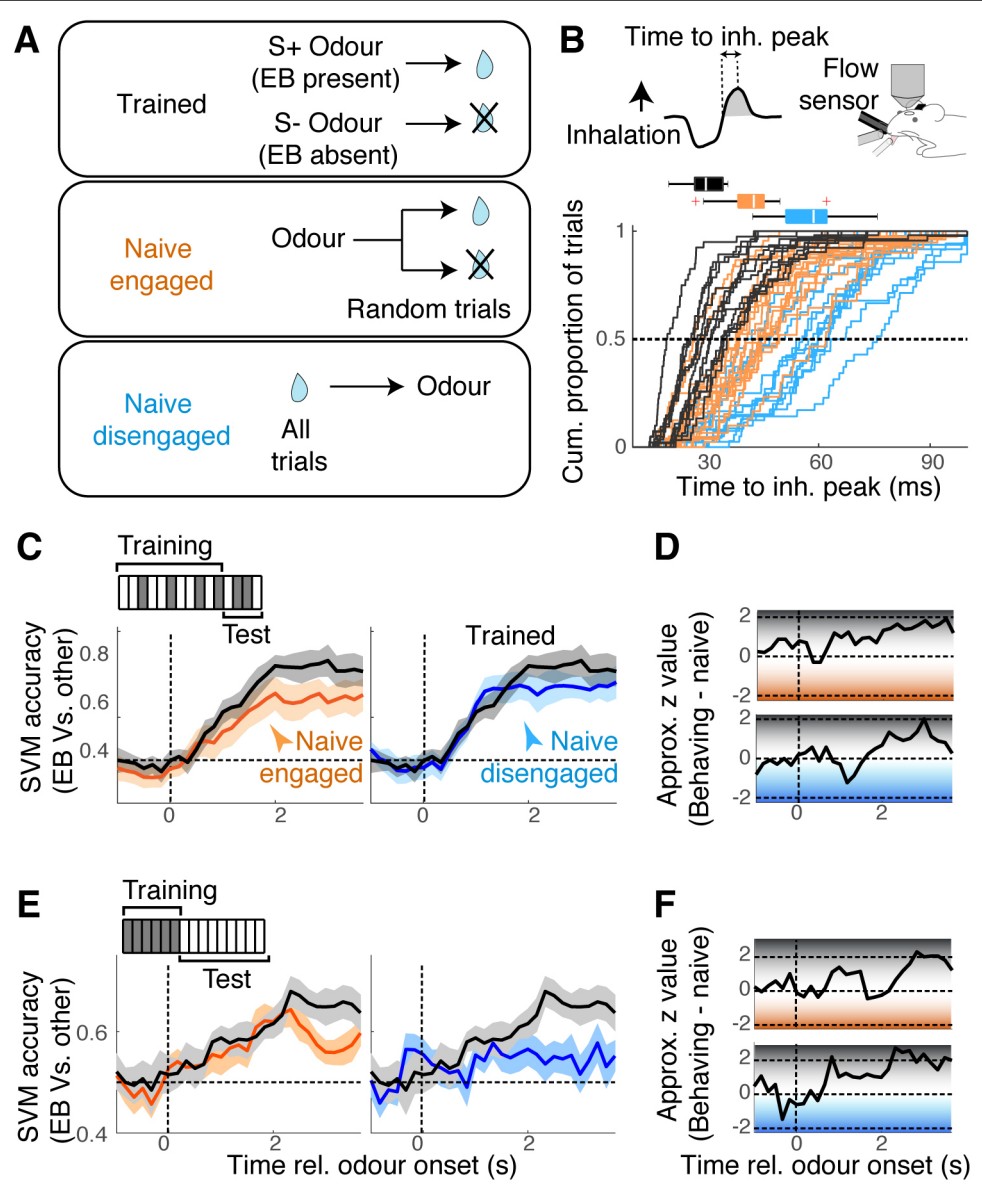

**Figure 6.** Mixture representation depends on behavioural states. (**A**) Reward contingencies used to achieve different behavioural states in awake mice. Top: mice were trained to discriminate between odours with ethyl butyrate (EB) vs. odours without EB. Middle: naïve mice received water reward 3 s after the onset of odour, on randomly selected trials. Bottom: naïve mice received water every trial, 15 s before odour onset. (**B**) Sniff patterns associated with trained and behaving mice (black), naïve and engaged mice (orange), and naïve and disengaged mice (light blue). Speed of inhalation during odour presentations shown. (**C**) Comparison of support vector machine (SVM) performance using data from behaving mice (black) vs. naïve engaged mice (left panel, orange), and vs. naïve disengaged mice (right panel, light blue). SVM was trained to discriminate responses to S+ vs. S- odours using randomly selected 80% of trials and tested on the remaining 20% of the trials. (**D**) Decoder performance for the results in (**C**) plotted using approximate z-values obtained from MATLAB implementation of Wilcoxon rank-sum test. (**E**) As in (**C**), but SVM was trained using responses to single odours, and tested on mixture responses. (**F**) As in (**D**) but for the results in (**E**). n = 13 fields of view, six mice for behaving, 17 fields of view, six mice for naïve, engaged, and 14 sessions, four mice for naïve, disengaged mice. See *Figure 6—figure supplement 1* for linearity of summation.

The online version of this article includes the following figure supplement(s) for figure 6:

**Figure supplement 1.** Further characterisation of awake behavioural states.

**Figure supplement 2.** Relationship between discriminability and linearity of mixture summation.

*Figure 6 continued on next page*

*Figure 6 continued*

**Figure supplement 3.** Mixture representation in tufted cells (TCs) is not affected by ipsilateral piriform inactivation.

accompanied by more linear summation. Second, the time course of decoder accuracy does not follow the linearity of summation. Indeed, in the data from behaving mice, the decoder performance peaked when mixture responses were most sublinear. This raises a question: is sublinear summation in mixtures detrimental or beneficial to discriminating mixtures of odours? Our simple simulation indicates that, generally, a sublinear mixture summation due to saturating influences limits the discriminability of mixture responses. Thus, dampened responses in awake animals bring some advantage by maintaining the mixture responses in the linear range. However, in the behaving mice, even when the responses became larger at a later phase, the saturating influence was less detrimental to discrimination. Here, decorrelated response patterns may play a more crucial role as this is known to enhance classifier or behavioural performances (*Bhattacharjee et al., 2019*; *Friedrich and Wiechert, 2014*; *Gschwend et al., 2015*; *Padmanabhan and Urban, 2010*). While the behavioural task described in this study is simple and animals make accurate decisions within 1 s (771 ± 97 ms) of stimulus onset, the slower mechanisms described here may be important when larger and more reliable responses are required for accurate decisions. So, in addition to sampling time (*Rinberg et al., 2006*), this phenomenon may be a part of mechanisms needed to solve more difficult tasks accurately (*Abraham et al., 2010*; *Wilson et al., 2017*).

Mechanistically, the ability to accurately analyse, or discriminate between, mixtures based on the knowledge of component is hypothesised to involve the piriform cortex, through learned representations (*Chapuis and Wilson, 2011*; *Grabska-Barwińska et al., 2017*). For tufted cells, however, the relevant sources of feedback could be other brain regions, such as the anterior olfactory nucleus (*Chae et al., 2020*). It will be an intriguing future investigation to identify the source and mechanisms of state-dependent mixture representations.

Perception of olfactory mixtures has long fascinated investigators. Mixtures of odours often have qualities that are different from those of the individual components. Accurate recognition of components is particularly hard for human subjects. For untrained subjects, olfactory mixtures that contain about 30 components tend to smell alike (*Weiss et al., 2012*). Even highly trained people like perfumers can only accurately identify individual components if unfamiliar mixtures contained no more than five components (*Poupon et al., 2018*). These demonstrate an ultimate limit in the analytical perception of olfactory mixtures. In addition, in non-expert humans and rodents alike, when the task is to look for a particular aroma in the mixture, the tendency is to falsely report the presence of the target odour in target-odour detection tasks (*Laing and Glemarec, 1992*; *Rokni et al., 2014*). In all cases, an extensive training for specific odours can improve the ability to detect the target odour, as seen in the case for sommeliers who routinely analyse key components in wines, which can contain several hundred component mixtures (*Ilc et al., 2016*). Thus, while mixture perception is highly context specific (*Rokni and Murthy, 2014*), training in specific odours seems to be key to improving on mixture analysis. With more complex mixtures, mechanisms within the OB may reach a limit, and also is unlikely to be the only mechanism that is used to solve the task. Elucidating what roles the primary sensory areas plays is a crucial step towards a mechanistic understanding of complex sensory processing.

## Materials and methods
### Animals

All animal experiments have been approved by the OIST Graduate University's Animal Care and Use Committee (protocol 2016-151 and 2020-310). *Tbx21^Cre^* (*Haddad et al., 2013*) and B6J. Cg-*Gt(ROSA)26Sor^tm95.1(CAG-GCaMP6f)Hze^*/MwarJ, also known as Ai95D (*Madisen et al., 2015*), were originally obtained from the Jackson Laboratory (stock numbers 024507 and 028865, respectively). *Tbx21^Cre^*::Ai95D mice were generated by crossing homozygous *Tbx21^Cre^* and Ai95D mice, resulting in heterozygous animals used for imaging experiments. C57Bl6J mice were purchased from Japan CLEA (Shizuoka, Japan) and were acclimatised to the OIST facility for at least 1 week before they were used for experiments. All mice used in this study were adult males (8–11 weeks old at the time of surgery).

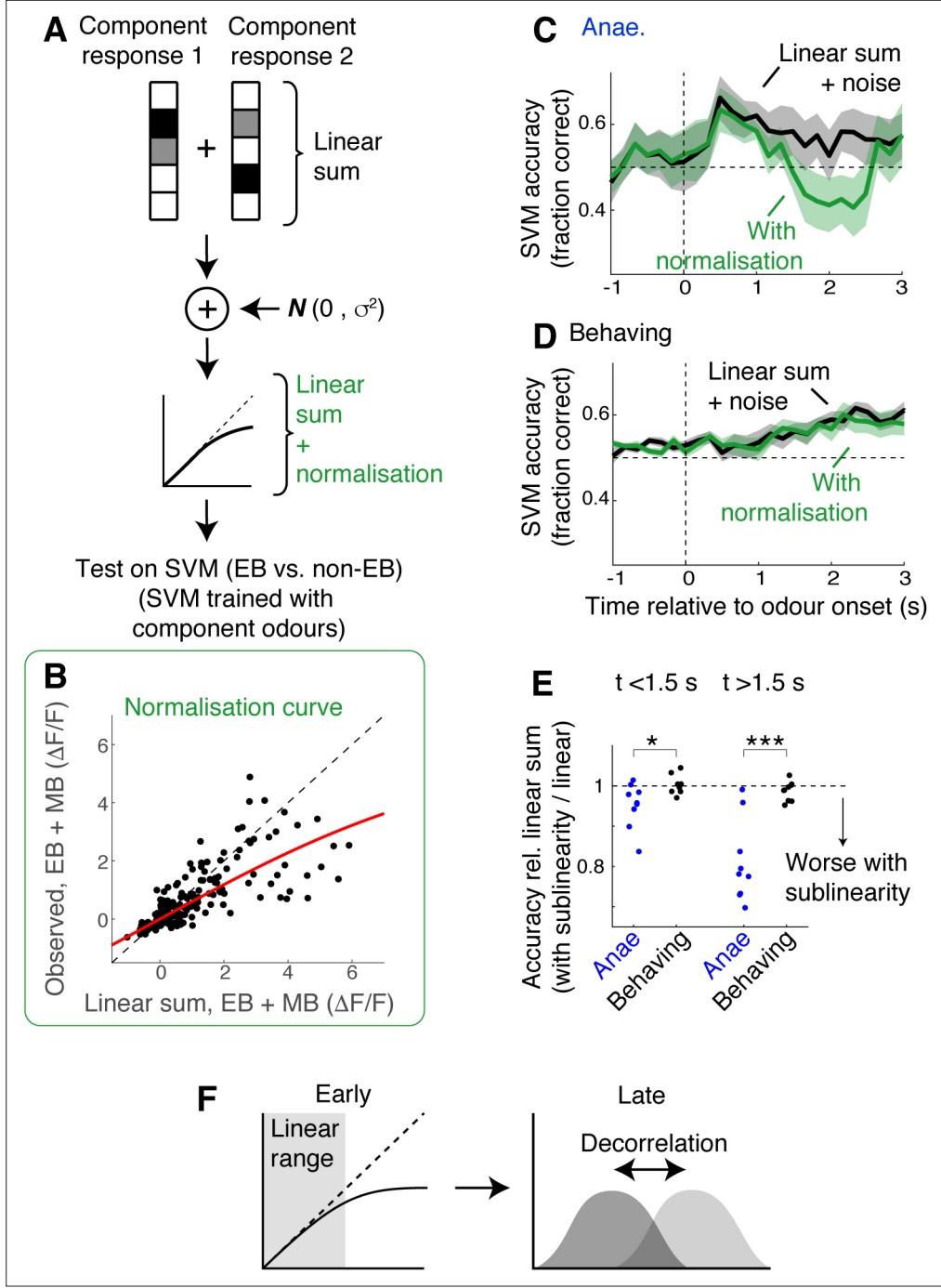

**Figure 7.** Simulation indicates data from behaving mice is less susceptible to normalising sublinearity. (**A**) Simulation approach. For each field of view, component responses were averaged across trials and summed to obtain simulated mixture responses. Subsequently, uncorrelated Gaussian noise was added, and the summed responses were further transformed using a normalisation model (*Mathis et al., 2016*; *Penker et al., 2020*). These were then used as test inputs on support vector machines (SVMs) that had been trained with single-odour responses. (**B**) Parameter extraction. Scatter plot shows observed ethyl butyrate (EB) + methyl butyrate (MB) mixture responses against linear sums of component responses (black dots). Red line shows the model fit used in the simulation. Data was from behaving mice, at 2 s after the odour onset. (**C**) Comparison of SVM performance using data from anaesthetised mice. Black line corresponds to linear sum + noise as inputs; green line corresponds to the results with additional normalising sublinearity. Mean ± SEM shown. (**D**) Same as in (**C**) but for data from awake, behaving mice. (**E**) Comparison of SVM performance. SVM accuracy for simulated mixture responses

*Figure 7 continued on next page*

*Figure 7 continued*

with sublinearity expressed as a fraction of accuracy obtained with simulated linear sum. p = 0.03 for early phase (0–1.5 s) and 8.13 * 10$^{-4}$ for late phase (1.5–3 s after odour onset); paired *t*-test. (**F**) Schematic of the finding: possible mechanisms of mixture representation in behaving mice may evolve over time. In awake, behaving mice, responses evoked in the olfactory bulb (OB) output neurons are dampened early on, remaining largely in the linear range of the normalisation curve. Over time, the responses become larger, making sublinearity more widespread, but a pattern decorrelation may make evoked mixture responses still discriminable.

## Olfactometry

A custom-made flow-dilution olfactometer was used to present odours. Briefly, custom Labview codes were used to control solenoid valves, and a flow controller (C1005-4S2-2L-N2, FCON, Japan) was used to regulate the rate of air flow. A pair of normally closed solenoid valves was assigned per odorant and used to odorise the air. These solenoid valves were attached to a manifold, such that a set of eight pairs had access to a common stream of air. To generate binary mixtures of odours while keeping concentrations stable, one odour from each manifold (lines A and B) was used. For single-odour presentations, odour was mixed with air that passed through an empty canister. The odorised air was directed towards the animal only when the solenoid valve closest to the animal (final valve) opened. Final valve was opened for a short time (0.1 s for experiments involving behavioural analysis only, and 0.5 s for all imaging experiments, to accommodate for slow respirations during anaesthesia), to avoid adaptation related to temporal filtering for high sniff frequencies associated with long odour pulses (*Verhagen et al., 2007*). Odours were presented at 1–5% of the saturated vapour. Total air flow, which is a sum of odorised air and the dilution air, was approximately 2 l/min, which was matched by the air that normally flows towards the animal. Inter-trial interval was approximately 40 s to purge airways with clean, pressurised air and to ensure that the flow controllers have stabilised before each odour presentation. All odorants were from Tokyo Chemical Industry (Tokyo, Japan), apart from EB (W242705), which was from Sigma-Aldrich. Product numbers were T0247 (ethyl tiglate), V0005 (methyl valerate), A0061 (acetophenone), A0500 (methyl anthranilate), B0757 (butyl butyrate), B0763 (MB), S0015 (methyl salicylate), S0004 (salicylaldehyde), T0248 (methyl tiglate), and A0232 (eugenol). Purity of all odorants was at least 98% at the time of purchase. Stock odorants were stored at room temperature in a cabinet filled with N$_2$ and away from light.

## Surgery

### Head plate implantation

All recovery surgery was conducted in an aseptic condition. 8–11-week-old male C57Bl6/J mice were deeply anaesthetised with isoflurane. The body temperature was kept at 36.5°C using a heating blanket with a DC controller (FHC, Bowdoin, USA). To attach a custom head plate about 1 cm in width weighing a few grams, the skin over the parietal bones was excised and the soft tissue underneath was cleaned, exposing the skull. The exposed skull was gently scarred with a dental drill, cleaned, dried, and coated with cyanoacrylate (Histoacryl, B. Braun, Hessen, Germany) before placing the head plate and fixing with dental cement (Kulzer, Hanau, Germany). For optical window implantation, adult, male *Tbx21$^{Cre}$*::Ai95D mice were deeply anaesthetised and underwent a window implantation procedure as previously described (*Koldaeva et al., 2019*). Briefly, after exposing the frontal bone, a craniotomy about 1 mm in diameter was made over the left OB and on the exposed dorsal surface, a cut piece of coverslip that snugly fit in the craniotomy, on the edge of the drilled bone, was gently pressed down, sealed with a cyanoacrylate and fixed with dental cement. Mice were allowed to recover in a warm chamber, returned to their cages, and given carprofen subcutaneously (5 mg/kg) for 3 consecutive days.

## Habituation and behavioural measurements

Water restriction began 2 weeks after the surgery. Mice went through 3 days of habituation to head fixation, one session per day for approximately 30 min, until mice learned to lick vigorously for water reward. Respiration pattern was measured by measuring the air flow just outside the right nostril by placing a flow sensor (AWM3100V, Honeywell, NC), and the data was acquired at 1 kHz. Lick responses were measured using an IR beam sensor (PM-F25, Panasonic, Osaka, Japan) that was part of the water port. Nasal flow, an analog signal indicating the odours used, lick signal, a copy of the

final valve, and water valve timing were acquired using a data acquisition interface (Power1401, CED, Cambridge, UK).

### Discrimination training

After habituation, the head-fixed mice were trained to associate a water reward with a target odour (EB). The reward was two droplets of water (10 μl each), which arrived 3 s after the onset of the final valve opening. The mice underwent single-odour discrimination training first until they generated anticipatory licks in response to EB presentations and correctly refrained from licking in response to other single odours. Once the overall accuracy was above 80% in at least one behavioural session, the mice went through the mixture detection task. A typical training session comprised roughly 150 trials, lasting about 1 hr. Rewarded trials comprised a third of all trials. Two-photon imaging took place once the mice performed at 80% accuracy or above in MB trials.

### Random association paradigm

After habituation, the head-fixed mice were presented with the same odour mixture stimuli as those that underwent the discrimination training. The water reward was delivered on randomly selected trials, 3 s after the onset of the final valve opening. One behavioural session was used to accustom the mice to the odours. Two-photon imaging commenced from the second behavioural session.

### Disengagement paradigm

After habituation, the head-fixed mice were presented with the same odour mixture stimuli as above. The water reward was delivered every trial, 15 s before the onset of the final valve opening, arriving in the middle of the inter-trial interval, which was 40 s to ensure thorough purging to clear the lines, as well as to stabilise the flow controllers. Two-photon imaging commenced from the second behavioural session.

### Odour mixture trial composition

Binary mixtures have been chosen due to the smaller number of possible odour combinations compared to more complex mixtures. However, even with 11 odours, there is a limit in the number of trials each head-fixed mouse can sample in a given session. We therefore decided to focus on the EB + MB mixtures. However, it was crucial that EB, MB, and EB + MB mixture is not presented too frequently to avoid possible adaptation, as well as mice becoming over familiar, especially in the EB-detection task, where the goal was not to train mice to remember specific odour combinations. Thus, we used a compromise paradigm where EB, MB, and EB + MB appeared every 10 trials.

### Data analysis

The data was analysed offline using custom MATLAB codes. To calculate the accuracy, the number of licks during 3 s from the final valve onset was measured for each trial. Threshold for an anticipatory lick was set to 2, thus the correct response for rewarded trials was two or more beam breaks, and the correct response for unrewarded trials was one lick or less during the response time window. To calculate the learning curve, the accuracy was expressed as the proportion of correct trials in a given block of 50 trials. *Time to inhalation peak*: to calculate the speed of inhalation, onset of inhalation and peak of inhalation was detected using Spike2 (CED, Cambridge, UK), using the built-in event detection functions. Briefly, inhalation peaks were detected using the 'rising peak' function. These events were used to search backwards in time for the inhalation onset when the flow signal crossed a threshold value. *Lick preference index*: to measure how well mice discriminated rewarded vs. unrewarded mixtures, anticipatory licking patterns for the two types of trials were compared using the following formula:

$$\text{Lick preference index} = (\text{Lick}_{\text{rewarded}} - \text{Lick}_{\text{unrewarded}})/(\text{Lick}_{\text{rewarded}} + \text{Lick}_{\text{unrewarded}}).$$

where $\text{Lick}_{\text{rewarded}}$ corresponds to the average number of anticipatory licks on rewarded trials, and $\text{Lick}_{\text{unrewarded}}$ corresponds to that for unrewarded trials.

## Imaging

A custom-made two-photon microscope (INSS, UK) with a resonant scanner was used to observe fluorescence from the OB of *Tbx21Cre::*Ai95D mice in vivo. 3D coordinates for imaged fields of view were recorded relative to the location of a reference, blood vessel pattern on the surface. Unless otherwise stated, imaging from somata was done relatively superficially, just below the glomeruli, thus mainly comprised TCs, which use firing rate modulation to represent odours (*Fukunaga et al., 2012*). Fields of view for glomerular and somatic levels were 512 μm × 512 μm, and 256 μm × 256 μm, respectively, and overlapped for awake and anaesthetised conditions, but fewer sessions took place under anaesthesia. In each trial, 400 image frames were acquired at 30 frames per second, with 200 frames before the final valve opening to obtain a steady baseline. Unless otherwise stated, the time window analysed for the odour-evoked responses was the first 1 s since the onset of final valve opening. For imaging under ketamine/xylazine anaesthesia (100 mg.kg$^{-1}$/20 mg.kg$^{-1}$ intraperitoneally), mice were kept on a warm blanket (FHC) to maintain the body temperature at 36°C.

## Ipsilateral muscimol infusion in the anterior piriform cortex

Animals were unilaterally implanted with a 26G cannula (10 mm length; C315GS-4/SPC, Plastics One). The cannula was inserted through a small craniotomy at a stereotactic coordinate (AP, 2.2 mm; ML, 2.4 mm relative to bregma) and advanced over 6.1 mm at 45° to target the anterior piriform cortex, and fixed with dental cement (Kulzer, Hanau, Germany). For infusion, 500 nl of 2 mM muscimol (M1523, Sigma-Aldrich, MO) was injected using a Hamilton microsyringe, at a rate of 100 nl/min approximately 10 min before the imaging started.

### Post-hoc verification of infusion cannula placement

Animals were infused with 500 nl DiI (Invitrogen, V22885), immediately transcardially perfused with phosphate buffer (225.7 mM NaH$_2$PO$_4$, 774.0 mM Na$_2$HPO$_4$, pH 7.4) and PFA (4%), dissolved in phosphate buffer. The heads (including the implants) were postfixed in PFA at 4°C for at 48–96 hr before extraction of the brains. Coronal sections of 100 μm thickness were cut on a vibratome (5100 mz-Plus, Campden Instruments, Leicestershire, UK) and counterstained using DAPI (D9542, Sigma-Aldrich). Images were acquired using a Leica SP8 confocal microscope using a ×10 (NA 0.40 Plan-Apochromat, Leica) objective. Images were taken at a resolution of 1024 × 1024 pixels per field of view (1163.64 × 1163.64 μm).

## Image analysis

Transients were extracted as follows. ROIs were manually delineated using an average frame from each imaging session. For glomeruli, ROIs were delineated by tracing the outer rim of congregated labelled processes surrounded by a dark region. Pixel values within each ROI were averaged to obtain a time series. Imaging sessions with motion artefacts and drifts were removed from analyses. For each transient, the baseline period was defined as 2 s preceding the final valve opening. Relative fluorescence change (ΔF/F) was calculated with respect to this baseline. Odour response period was 1 s (30 frames) starting at the onset of the final valve opening, unless otherwise stated. Awake mice tended to adjust sniff patterns, so the final valve opening was not triggered by nasal flow. Masking index for the glomerular level was calculated as previously described (*Rokni et al., 2014*). Briefly, responses to single odours were obtained from glomeruli of anaesthetised mice. In each field of view, only the glomeruli responsive to the target odour (EB) were analysed. To obtain glomeruli evoked by the target odour, the evoked response amplitudes were converted into z-scores. Glomeruli that responded to EB with z-scores higher than 2 were considered. The masking index was the average overlap in the evoked response, where the maximum value for each glomerulus was 1. For each odour, responses were averaged over three or more trials. Mean and standard error of the mean are shown in figures, unless otherwise stated. Boxplots were constructed using the MATLAB function *boxplot* and show the median, 25th and 75th percentiles. Outliers are shown with red crosses. The experiments were not done blindly since the stimulus-reward contingency was visible to the experimenter. However, the olfactometer performance, age and sex of the mice, and analysis codes used were the same for all conditions.

*Fractional deviation from linearity* was calculated as,

$$(R_{observed} - R_{linear\ sum})/\mid R_{linear\ sum}\mid,$$

where $R_{observed}$ is the mean response amplitude for observed mixture (e.g. response to EB + MB mixture), $R_{linear\ sum}$ is the trial average linear sum of component responses (e.g. EB response + MB response). Median fractional deviation was obtained from each field of view. For testing the effect of trial-by-trial variability, Gaussian noise was generated using the MATLAB function *normrnd* with the mean set to 0, and added to responses from each trial, before the fractional deviation was calculated.

*Deviation from linearity* was the difference between an observed mixture response amplitude and a linear sum of components normalised by the joint standard deviation:

$$(R_{observed} - R_{linear\ sum})/(s.e.m._{observed} + s.e.m._{linear\ sum})$$

*Discriminability index* based on correlation coefficient was calculated in the same way as *Discr*, described in *Bridgeford et al., 2021*, except that the distance measure used was 1 – Pearson's correlation coefficient, instead of the Euclidean distance.

## Decoding analysis

SVMs were trained using the MATLAB function *fitclinear* with linear model intercept ('*FitBias*') set to 0. Test datasets were used to predict the trial types using the MATLAB function *predict*. Signals were averaged over 30 frames (~1 s) for each time point to obtain the time course.

## Simulating the effect of normalising sublinearity on discriminability

SVMs were trained as above using the single-odour responses only. Then simulated mixture responses were constructed as follows. Single-odour responses were averaged over trials. Combinations of two averaged responses were added linearly, and uncorrelated Gaussian noise (mean = 0) was added. The noise term comprised two components: fixed, baseline standard deviation (0.2 for anaesthetised data, and 2 for awake, behaving case) and standard deviation that scaled with response amplitude by taking the slope of a linear regression of the observed data. In one test, linear sum with added noise was used as inputs to the above SVMs. In a second set of tests, these simulated responses were further passed through a normalising sublinearity and tested on the same SVMs. The equation of normalisation (*Penker et al., 2020*) was

$$R_j^* = R_{max}\left(\frac{2}{1+e^{-s.R_j}} - 1\right)$$

where $R_j$ represents the linear-sum amplitude of the jth neuron, and $R_j^*$ is the normalised response amplitude of jth neuron. Parameters $s$ and $R_{max}$ were obtained by fitting the data to observed mixture responses (on data from behaving animals at t = 2 s after odour onset). $R_{max}$ was 6 and $s$ was 0.2.

## Acknowledgements

We thank Yu-Pei Huang and OIST's Animal Resource Service staff for their dedicated assistance, the members of Andreas Schaefer's group and Kevin Franks' group for comments on earlier versions of the work, and Adam Mago, Xiaochen Fu, and Josefine Reuschenbach for comments on the manuscript. This work was supported by the OIST Graduate University.

## Additional information

### Funding

| Funder | Grant reference number | Author |
| --- | --- | --- |
| Okinawa Institute of Science and Technology Graduate University | | Aliya Mari Adefuin Sander Lindeman Janine K Reinert Izumi Fukunaga |

| Funder | Grant reference number | Author |
|--------|------------------------|--------|

The funders had no role in study design, data collection and interpretation, or the decision to submit the work for publication.

## Author contributions
Aliya Mari Adefuin, Conceptualization, Data curation, Formal analysis, Investigation, Writing – review and editing; Sander Lindeman, Janine Kristin Reinert, Investigation, Supervision, Writing – review and editing; Izumi Fukunaga, Conceptualization, Formal analysis, Funding acquisition, Investigation, Supervision, Writing - original draft

## Author ORCIDs
Sander Lindeman ![ORCID] http://orcid.org/0000-0002-2965-744X
Janine Kristin Reinert ![ORCID] http://orcid.org/0000-0002-1495-7431
Izumi Fukunaga ![ORCID] http://orcid.org/0000-0003-1860-5377

## Ethics
All procedures described in this study have been approved by the OIST Graduate University's Animal Care and Use Committee (Protocol 2016-151 and 2020-310).

## Decision letter and Author response
Decision letter https://doi.org/10.7554/eLife.76882.sa1
Author response https://doi.org/10.7554/eLife.76882.sa2

# Additional files

## Supplementary files
• Transparent reporting form

## Data availability
The data generated in the study is available in Dryad Digital Repository at 10.5061/dryad.p2ngf1vrh. The files consist of individual data to compare linear sum vs. observed mixture responses (300 - 1000 ms after odour onset).

The following dataset was generated:

| Author(s) | Year | Dataset title | Dataset URL | Database and Identifier |
|-----------|------|---------------|-------------|-------------------------|
| Fukunaga I | 2022 | Figure 6C | https://dx.doi.org/10.5061/dryad.p2ngf1vrh | Dryad Digital Repository, 10.5061/dryad.p2ngf1vrh |

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
