## [Editor Report]

This study provides a strong evidence supporting that odour mixture interactions are more linear during awake animals compared to anaesthetised conditions, contrary to the previous notion that odour mixture interactions are sublinear – the conclusion obtained in anaesthetised animals. With their revisions, the authors have done new simulations to clarify how linear mixture interactions affect odour discriminability and interact with other factors (e.g. decorrelations). This new analysis provides a reasonable explanation as to when linearity helps improve discriminability and clarify the significance of sublinear interactions.

---

## [Decision Letter]

**Decision letter after peer review:**

[Editors’ note: the authors resubmitted a revised version of the paper for consideration. What follows is the authors’ response to the first round of review.]

Thank you for submitting the paper "State-dependent linearisation of mixture representations by the olfactory bulb" for consideration by *eLife*. Your article has been reviewed by 3 peer reviewers, including Naoshige Uchida as the Reviewing Editor and Reviewer #1, and the evaluation has been overseen by a Reviewing Editor and a Senior Editor. We are sorry to say that, after consultation with the reviewers, we have decided that this work will not be considered further for publication by *eLife*.

Summary:

The reviewers agreed that this study addresses an important question: how complex odors are processed in the olfactory bulb, more specifically, whether mixture components are represented as a linear sum of components or sublinear interactions occur as suggested by previous studies. The results are potentially interesting, but the reviewers have raised substantive concerns.

1. The reviewers were not convinced by the analysis based on the fractions of sublinear versus linear interactions. The authors report the fraction of non-linear interactions as defined by the activity showing a deviation from linearity using a statistical criterion (> 2 standard deviation). The reviewers thought the authors need to perform more analysis by considering trial-to-trial variability, magnitude difference, and response saturation etc.. Each reviewer made useful suggestions (see below).

2. Odor responses in awake conditions are generally weaker than those under anesthesia. It is, therefore, unclear whether the overall discrimination is better during awake condition. It is important to perform decoding analysis using a classifier trained using odor responses in each condition (not a classifier built using predicted responses).

3. The reviewers had somewhat different opinions on overall significance of the work. One reviewer thought that the difference in awake and anesthetized conditions, if shown convincingly by addressing the above points, is important because previous studies have predominantly reported sublinear interactions. One reviewer, however, thought that the difference between awake and anesthetized conditions is not sufficiently novel, because many other differences have already been reported. One reviewer thought that further mechanistic insights would be important.

Overall, strong demonstrations of the above point 1 and 2, and/or further mechanistic insights would need to be considered for *eLife*. If the authors can address the above issues, we would be willing to reconsider a resubmitted version but without guarantees concerning publication.

*Reviewer #1:*

Adefuin and colleagues examined the interaction between components of binary odor mixtures in odor responses in mice. The authors used two-photon calcium imaging from the soma and apical dendrites of mitral/tufted cells in the olfactory bulb. Odor responses were measured in various conditions: under anesthesia (ketamine/xylazine), while well-trained mice were engaged in an odor discrimination task, or disengaged. The authors first show that mixture components interacted sublinearly in a large fraction of mitral/tufted cells (46%; Figure 6D) consistent with previous studies. However, when odor responses were measured in awake animals, very few mitral/tufted cells showed sublinear responses at soma (8-9%; Figure 6D). Interestingly, sublinear interaction was evident in apical dendrites of mitral/tufted cells (45%). Whether mixture components are represented linearly or not in the olfactory system is an important question, related to the animal's ability to identify or segment mixture components. Somewhat contrary to previous studies, this study demonstrates largely linear interactions. Furthermore, this study compares various behavioral conditions. These results are important and of interest to those who study sensory systems. I have a few concerns regarding data analysis.

1. Non-linear interactions are detected by the activity showing a deviation from linearity greater than 2 standard deviations. Using this criterion, non-linear interactions might decrease if the trial-by-trial activity becomes more variable. This is concerning because the activity might be less variable in the anesthetized condition, and the reduction in sublinear interactions in awake conditions may be due to a general increase in response variability during awake. Can the authors exclude the possibility that the decrease in sublinear interactions is merely due to an increase in response variability in the awake conditions. This issue also applies to the comparison between apical dendrites versus soma; are the signals in apical dendrite less variable (maybe due to some averaging across dendrites from multiple cells; see the following point 5)?

2. Related to the above issue, it would be useful to analyze the difference between conditions using different metrics to fully understand what really are different between conditions. The scatter plots shown in various figures do not show drastic differences between awake and anesthetized conditions, as might be indicated by the percent of sublinear responses. It would be useful to characterize the magnitude of sublinear/supralinear effects. For example, one can calculate a fractional change in the mean response. Does this measure show consistent difference between awake and anesthetized conditions?

*Reviewer #2:*

This study addresses how complex stimuli are represented in neural responses. This is particularly relevant to olfaction because the vast majority of stimuli are complex mixtures that perceptually, are not easy to decompose into parts. Nonetheless, the ability to discern a relevant odor from background odors is essential. This process is easier when neural responses to mixtures reflect the linear sum of the responses to the individual components. The main conclusion of this study is that the linearity of olfactory bulb responses to two-component mixtures increases awake versus anesthetized states. The authors provide some evidence to support this claim. However, this could be better quantified and there is a temporal aspect of linearization that is not addressed. Perhaps the most interesting aspect of the study is the difference in linearity between the dendrites and the somata of the mitral/tufted cells. But a statistical analysis of this finding was not evident. Overall, a mechanistic or functional approach to understanding these findings is lacking. The differences linearity between the anesthetized and awake are simply explained by response saturation anesthetized animals. There are hints at mechanism by which linearity is supported in the OB with comparisons between soma and dendrite, but these are not well developed. There is a model that addresses the functional significance of linearity, but this is only supplemental and not well described.

1) The quantification of linearity is incomplete. The deviation from linearity is a fine metric but just reporting the change in fraction of ROIs that show deviation from linearity in the sublinear direction is insufficient. There are many ROIs that show supralinear summation as well. Moreover, it is not clear where the cut-off is between linear and non-linear responses. There are statistical analyses that can be done on proportional data to determine if the proportions of non-linear ROIs are statistically different in the anesthetized and awake states. These should be done.

2) Another approach is to compare the slope of the lines fit to data in the Observed vs. Linear sum plots and to determine how much these deviate from 1, and if this is statistically significant. These slopes could also be compared between states.

3) It is also necessary to be clear about which timepoint is being analyzed. It is apparent from figure 4 that the linearity of the population of ROIs evolves with time, becoming most linear 3 sec after odor onset. This temporal evolution of linearity is not addressed at all. But could provide some insight into mechanism.

4) There is no statistical analysis of the dendrite versus soma data. The figure is not very convincing of differences. Why wasn't the temporal progression of responses shown for the dendrite versus soma data?

5) Linearity seems very correlated with the strength of the response. Sublinearity increases with strength suggesting saturation is a major contributing factor. Conversely, supralinear responses are more apparent with weaker component responses. It is well known that OB responses are much stronger in anesthetized animals. The deltaF/F values are greater in the anesthetized state compared to the awake state. So, the differences between the states may be trivially explained by saturation (as shown in Supplemental Figure 6). This doesn't really say anything about how the OB linearizes responses. However, the temporal evolution of linearization could speak to other mechanisms such as adaptation, recruitment of inhibition, descending control etc. Analyses of the dendrite versus soma signals over time could give insight as to where/when these mechanisms may be occurring. I think this was attempted in the supplement to figure 7, but the figure does not have sufficient explanation to figure out what is going on.

6) There was a model of how linearity might function in odor detection, but this is not well described or motivated in the text.

7) Overall, all the supplemental figures and data are essential to the study. Many of my concerns are somewhat addressed by these additional analyses. But they are not well explained. These should be integrated into the main figures of the paper and adequately described in the Results section.

*Reviewer #3:*

Adefuin et al. use multiphoton imaging of M/T cell responses to investigate whether neuronal representations of binary mixtures can be explained as a sum of the components. The current view in the field (built largely from studies in anesthetized animals), is that mixture summation is non-linear and increases with the degree in glomerular response overlap elicited by the components. The authors reproduce these results and ask whether the same phenomenon is observed in the awake state, in particular when the animals are engaged in an odor discrimination task. Unlike in the anesthetized state, the authors find that mixture representations are linear in the awake brain. They use a series of systematic behavioral paradigms to show that the observed linearity in the awake state (compared to anesthetized) is not dependent on task engagement (reward is given randomly, post-odor) or stimulus relevance (reward is given before odor). While the experiments are well done and the data is presented clearly, I have several major concerns about the interpretation of their results.

1) Given the data the authors present, it is unclear if one can conclude that the olfactory system is more or less linear in the awake state compared to the anaesthetised one. What seems to change most across the awake vs. anesthetized state is the response amplitude. Responses appear to be ~3x smaller in the awake mice. In the anesthetized state, non-linearity seems most apparent for large response amplitudes (>5 dF/F) with mixture responses being sub-linear, most likely due to saturation effects. The authors themselves do an analysis in Figure 6 – supplement 1 to show that most of the observed non-linearity in the anesthetized animals can be explained away after accounting for amplitude normalisation. The authors use this analysis to comment that the level of linearity is the same across all the three awake states, but the same figure shows that it is in fact the same even for the anaesthetized state.

To put it differently, it is indeed true from the authors data that the OB response gain is significantly lower in the awake state, but it is unclear if the summation is more linear if measured at similar response amplitude regimes in both awake and anaesthetised mice.

2) The authors argue that keeping response amplitudes small in the awake brain prevents sub-linear summation and therefore may lead to better mixture decomposition. They do a decoding analysis in anaesthetised mice to show that linear mixture representations (instead of using observed sub-linear representations) make odor classification easier. However, I find this analysis uninformative and misleading. It is no surprise that the decoders trained on single odor representations should perform better (or equivalent) when using linear sums as input instead of observed sub-linear representations. The authors use this observation to suggest that this mechanism aids discrimination ability in the awake state. However, given that even the single odor responses are much weaker and noisier in the awake state, it is likely that even the single odor discrimination ability is poorer in the awake state. By the same logic, mixture decomposition might be also much poorer in the awake brain than the anesthetized brain, even though summation is more linear, just because responses are weaker and noisier. In my opinion, the authors should compare decoding accuracy across awake vs. anesthetized responses if they want to assert that linearisation of responses in the awake brain leads to easier decomposition. Because otherwise, while linearisation in principle can aid decomposition, at least in the form that the authors observe here, it may come at a high cost on signal-to-noise ratio which would undo the gain that linearity provides, in principle, for discrimination.

3) At a more philosophical level, to this Reviewer, it is unclear if anesthesia vs. awake state difference in response should constitute the main focus of the manuscript. The authors explore summation properties under four different brain states, one of which is anaesthesia (also least behaviorally relevant). In three out of four states, they observe that summation is linear. In the fourth (anaesthesia), they observe that summation is sub-linear, but this happens at much larger response amplitude regimes compared to the three awake states sampled, presumably due to saturation. To me, it seems that the Authors here show that mixture summation in the OB, is largely independent of brain state since it is unaffected by whether the animal is task engaged or motivated etc.

4) It is unclear how to interpret the dendritic imaging comparison. First, the dendritic signal is pooled across many cells. If any of the cells that are being pooled shows sub-linearity, the pooled population response will look sub-linear, albeit less so than at the single cell level. Second, again like for the anesthetized vs. awake comparison, there is a discrepancy in response amplitudes – dendritic responses are ~2x stronger than the somatic responses and sub-linear summation would be more apparent as one approaches the saturation regime. Third, dendritic responses pool both mitral and tufted, while the somatic data the authors present is predominantly from tufted cells.

[Editors' note: further revisions were suggested prior to acceptance, as described below.]

Thank you for resubmitting your work entitled "State-dependent representations of mixtures by the olfactory bulb" for further consideration by *eLife*. Your revised article has been reviewed by three peer reviewers, including Naoshige Uchida as the Reviewing Editor and Reviewer #1, and the evaluation has been overseen by Gary Westbrook (Senior Editor). The manuscript has been improved but there are some remaining issues that need to be addressed, as outlined below:

The authors have performed new data analysis and revised the manuscript. Overall, all reviewers thought that these changes have greatly improved the manuscript. In particular, the new metric to characterize linearity of mixture interactions -- median fractional change -- provides a more quantitative information, and the main conclusion has been strongly supported by this new method. The decoding analysis now shows that the odor information remains at least as informative in awake conditions as in anesthetized conditions, despite overall reduced magnitude of response in awake conditions. The new analysis also shows that discriminability does not necessarily correlate with the extent of linearity in mixture summation.

Although the manuscript has greatly improved, there are some remaining issues. Specifically, Reviewer 2 and 3 have raised concerns that need to be clarified. In particular, Reviewer 3 has raised the issue regarding the dissociation between discriminability and linear mixture summation. The functional significance of linear mixture summation, or the limitation thereof, needs to be discussed more carefully, and it would be useful to include some discussion on potential causes of increased discriminability in awake conditions. In addition, Reviewer 2, raised several points about clarifying and adjusting statements in the abstract and the text of the manuscript, and analyzing the temporal dependence of decoding accuracy to the instantaneous level of non-linearity in response when comparing across different brain states. We would like you to respond to these remaining issues.

*Reviewer #1:*

The authors have addressed my previous concerns by reanalyzing the data and revising the manuscript. Importantly, the authors now provide a more convincing metric for linearity of mixture interaction using the median fractional change over the population. I am more convinced by this metric which supports that mixture interaction is more linear in awake conditions. This new analysis addresses my most important concern. A new analysis also finds that decoding or behavioral ability to detect a mixture component did not correlate with the linearity of mixture interactions in odor responses. This finding, in a sense, makes the significance of the finding of linearity in odor responses a little unclear. On the other hand, I think that the demonstration that odor decoding does not benefit from linear mixture interaction has an important message to the field as one might assume that it would be the case. It would be more useful if the authors can demonstrate this point (odor decoding does not benefit from linear mixture interaction) in a more abstract form without relying on the data -- can we take this conclusion as a general result, or does it rely on some particular features in the data. A simple modeling might help address this issue. Although such demonstration is not required, if the authors can address this in a short period of time, it would enhance the manuscript significantly, although this point is a little secondary compared to the main point (linearity of odor mixture interaction).

Overall, I am satisfied by the more convincing demonstration of linearity of mixture interaction. Because sublinearity has been emphasized in the previous literature, the results presented in this manuscript provide an important result.

*Reviewer #2:*

I commend the authors for re-structuring the manuscript and performing additional analyses to address the concerns raised. These changes have indeed improved the manuscript substantially.

However, I still have concerns about the disparity in response amplitude and dynamics across the different states and how that may affect the interpretation of the results. I think these can be addressed by further clarifying and editing the statements in the abstract and rest of the manuscript before publication.

1. The authors use the new metric – Fractional deviation – to compare differences in the degree of linearity between the anesthesia and awake states. While this metric normalizes deviations from linearity with the response amplitude, it does not correct for the overall difference between the means of the two response amplitude distributions – in anesthesia versus awake states. If majority of the responses during the awake state are smaller than during anesthesia, and the amount of non-linearity scales with response amplitude – irrespective of the metric chosen, there will be overall more non-linear instances in anesthesia than in the awake state. This does not imply that large amplitude responses will sum 'more' linearly in the awake state. To this reviewer, an adequate comparison to test this possibility is sub-sampling the response distribution under anesthesia and comparing differences in non-linearity between responses of similar amplitudes. Do similar amplitude responses tend to sum more, or less linearly in awake versus anaesthetized states?

I think that the observation that, in response to the same stimuli, at the level of mitral and tufted cells, are kept in the linear regime of the system is insightful and would be important to document. It would enable the field to move forward. In my opinion, one way for the authors to clarify this matter is to show both the normalized and un-normalized data in the manuscript and simply state that in the awake state, responses to the same stimuli tend to be smaller in amplitude than under anesthesia, and, therefore, remain in the linear summation regime (i.e. do not get to reach high amplitudes that would sum up in a more non-linear fashion). This could be due to, for example, to stronger gain control mechanisms in the awake state, etc.

2. The authors added analysis to compare the discriminability between the anesthesia and awake state. While it is reassuring that discriminability is decent even with the smaller amplitude responses in the awake state. I find that the overall interpretation of the results could potentially lead to confusion in the readers' minds. Therefore, I propose to clarify by adding a few explanatory statements, and adjusting some of the statements in the abstract.

If I understand correctly, the authors use a 0.5s pulse of odor and find that discriminability during the first second or so, is comparable (for training an testing on random selection of single odor responses and mixture responses) or better (for training on single odors, testing on mixtures) during anesthesia. In the later phase (>1.5 s), awake responses appear to do better at discriminability. I am not sure how to interpret this finding. Could this simply be related to differences in response amplitudes and temporal dynamics? If so, stating this, would be very useful to bring clarity to the message. From the examples in Figures 1,2,4 it appears that responses during anesthesia peak early and decay back to baseline within 1 second from odor OFF (so, mostly back to baseline at 1.5 s from odor ON), whereas responses during the awake state have longer latencies and peak later. Therefore, the time-course differences in responses might explain the time-course difference in discriminability. It would be important to state that these differences in response dynamics across the two states may be the basis for the time differences in discriminability.

From a behavioral point of view, it is unclear how to interpret these time-courses, discriminability peaks much later than behaviorally relevant time-scales in the awake state (<1.5s for the odor concentration used, 1-5%), and while odor identities can be decoded in anesthesia – animals don't make decisions under KX. When comparing the discriminability between naïve engaged, it would be important to clarify whether these differences arise early, at behaviorally relevant time-scales, or only appear in the later phase of the responses (>1.5 s) from odor onset and adjust the statements in the abstract accordingly. A side by side comparison would clarify this point.

3. Discriminability as a function of linearity of summation. It is unclear what the authors mean by 'the time course of decoding accuracy did not correlate with the linearity of summation'. The authors do not quantify linearity as a function of time, or explicitly analyze the temporal dependence of decoding accuracy with the instantaneous level of non-linearity. To support the above-mentioned sentence, such analysis would be necessary. Additionally, while this is a nice result (that decoding accuracy is not a direct correlate of linearity), it dampens the significance of assessing differences in linearity between various brain states.

*Reviewer #3:*

The authors have made a good attempt to address concerns raised in the first review. They have reanalyzed the data and used an alternate metric- mean fractional deviation, to evaluate response linearity. Importantly their original result still holds, M/T responses are more linear in the awake state than anesthetized. There is still some issue with stating the importance of this finding. It still reads like a non-finding from a functional perspective. Linearity only seems to be helpful with respect to discrimination in a time frame (>2s) that is after the animal has already started licking. But it is also important to note that despite a reduction in discrimination in early odor responses compared to anesthetized this does present a problem at the behavioral level. Particularly for a simple task such as the one used here. The longer time frame maybe important for more difficult tasks. Also, a potentially underestimated finding is the overall reduction in correlation between M/T cells in the awake state (Figure 5B). Such a reduction in correlation decreases redundancy which may enhance coding (Padmanbhan and Urban, 2010). These points should be emphasized more.

---

## [Author Response]

Summary:The reviewers agreed that this study addresses an important question: how complex odors are processed in the olfactory bulb, more specifically, whether mixture components are represented as a linear sum of components or sublinear interactions occur as suggested by previous studies. The results are potentially interesting, but the reviewers have raised substantive concerns.1. The reviewers were not convinced by the analysis based on the fractions of sublinear versus linear interactions. The authors report the fraction of non-linear interactions as defined by the activity showing a deviation from linearity using a statistical criterion (> 2 standard deviation). The reviewers thought the authors need to perform more analysis by considering trial-to-trial variability, magnitude difference, and response saturation etc.. Each reviewer made useful suggestions (see below).2. Odor responses in awake conditions are generally weaker than those under anesthesia. It is, therefore, unclear whether the overall discrimination is better during awake condition. It is important to perform decoding analysis using a classifier trained using odor responses in each condition (not a classifier built using predicted responses).3. The reviewers had somewhat different opinions on overall significance of the work. One reviewer thought that the difference in awake and anesthetized conditions, if shown convincingly by addressing the above points, is important because previous studies have predominantly reported sublinear interactions. One reviewer, however, thought that the difference between awake and anesthetized conditions is not sufficiently novel, because many other differences have already been reported. One reviewer thought that further mechanistic insights would be important.Overall, strong demonstrations of the above point 1 and 2, and/or further mechanistic insights would need to be considered for eLife. If the authors can address the above issues, we would be willing to reconsider a resubmitted version but without guarantees concerning publication.

We thank you for your thoughtful and helpful comments on the earlier version of our manuscript. We are pleased to re-submit what we believe to be a substantially improved version, with more robust analyses, and more insights, manifesting in new main figures, and 3 new supplemental figures.

First, we re-analysed the data, taking into account the trial-by-trial variability in the awake data (Distilled Pointed #1 above). We revised how we describe the mixture summation, which is now more robust to differences in noise and amplitudes across different conditions. Specifically, we decided to use *median fractional deviation*. As the name suggests, it is the median of fractional deviation, which we demonstrate in the revised Figure 4 to be robust to noise. In addition, we no longer use an arbitrary cut-off to classify the types of summation. We still find the mixture summation to be more linear in awake animals in the early phase of the response.

Following the second suggestion (Distilled Point #2), we also carried out (1) an analysis of trial-bytrial variability and (2) analyses using discriminant decoders.

We found, as the reviewers noted, that the trial-by-trial variability is greater in the awake mice (now included in the revised Figure 5). This is reflected in the lower correlation between S+ odour responses (i.e., within-class correlation) in this condition, compared to the anaesthetised case. However, we also observed that the across-class correlation (S+ vs. S- responses) is lower in the awake case i.e., odour patterns across classes are more decorrelated.

Since, as the reviewers noted, it was unclear how these factors affect the ability to detect ethyl butyrate in mixtures, we carried out a decoder analysis. We found that discriminant decoders (SVMs) using data from behaving mice performed equally well, or even better than decoders using the anaesthetised data during the late phase (revised Figure 5).

Interestingly, we also observed that the time course of decoder accuracy does not match the linearity in summation, making it unlikely that the linearity is critically needed for mixture analysis. This is noted in the following place.

Abstract (lines 19-21): “…decoding analyses indicated that the data from behaving mice was able to encode mixture responses well, though the time course of decoding accuracy did not correlate with the linearity of summation”. Result (lines 200 – 203): “Curiously, the time course of decoder accuracy did not correlate with the linearity of mixture summation. Overall, the result suggests that, despite the substantial differences in the amplitudes and variability of responses, M/T cells in behaving mice are able to encode the presence of the target odour well.”

Regarding the data from awake mice with different training and engagement levels, a decoder analysis revealed a state-dependent difference, where data from disengaged mice performed particularly poorly, suggesting that mixture responses may be subject to modulation by behavioural contexts (revised Figure 6). Again, the decoder accuracy did not depend on the linearity of summation.

In terms of significance (Distilled Point #3), we understand the difficulty the reviewers may have faced with the previous version. However, as one reviewer recognises, prior to our investigation, study after study demonstrated widespread mixture suppressions, using anaesthetised animals. It is important for the field to know that this is not the case in awake animals. That is, while our observation indeed falls into the general category of “awake vs. anaesthesia differences”, it is still important for the field to understand how mixtures are represented in the primary olfactory area. Further, currently, whether or not nonlinear summation (“interaction between mixture components”) is detrimental to the analytical perception of odour mixtures is still under debate, with a number of classical studies suggesting a detrimental relationship (Bell et al., 1987; Laing, 1994), while more recent studies suggest otherwise, i.e., antagonism could be beneficial to the analytical perception (Reddy et al., 2018). Our observation that a behavioural improvement is not accompanied by a greater linearisation, and our (new) observation that the decoder performance does not correlate with linearity, should be relevant.

In summary, we believe that the revised manuscript addresses the major concerns and brings insights on how mixtures are represented in the primary olfactory area. We are grateful for your time and for your helpful comments, which have greatly improved our manuscript.

Please find our point-by-point responses below.

Reviewer #1:Adefuin and colleagues examined the interaction between components of binary odor mixtures in odor responses in mice. The authors used two-photon calcium imaging from the soma and apical dendrites of mitral/tufted cells in the olfactory bulb. Odor responses were measured in various conditions: under anesthesia (ketamine/xylazine), while well-trained mice were engaged in an odor discrimination task, or disengaged. The authors first show that mixture components interacted sublinearly in a large fraction of mitral/tufted cells (46%; Figure 6D) consistent with previous studies. However, when odor responses were measured in awake animals, very few mitral/tufted cells showed sublinear responses at soma (8-9%; Figure 6D). Interestingly, sublinear interaction was evident in apical dendrites of mitral/tufted cells (45%). Whether mixture components are represented linearly or not in the olfactory system is an important question, related to the animal's ability to identify or segment mixture components. Somewhat contrary to previous studies, this study demonstrates largely linear interactions. Furthermore, this study compares various behavioral conditions. These results are important and of interest to those who study sensory systems. I have a few concerns regarding data analysis.

Thank you for your helpful review, and for recognising the relevance our work. We hope that the reviewer finds the our point-by-point responses satisfactory.

1. Non-linear interactions are detected by the activity showing a deviation from linearity greater than 2 standard deviations. Using this criterion, non-linear interactions might decrease if the trial-by-trial activity becomes more variable. This is concerning because the activity might be less variable in the anesthetized condition, and the reduction in sublinear interactions in awake conditions may be due to a general increase in response variability during awake. Can the authors exclude the possibility that the decrease in sublinear interactions is merely due to an increase in response variability in the awake conditions. This issue also applies to the comparison between apical dendrites versus soma; are the signals in apical dendrite less variable (maybe due to some averaging across dendrites from multiple cells; see the following point 5)?

Thank you for raising this valid point and for suggesting alternative analyses. We agree that the index we used previously is susceptible to noise, and not appropriate for comparing two datasets with different trial-by-trial variability. To quantify the deviation from linear sum more robustly, we now use the “Median fractional deviation”, which expresses a deviation from the linear sum as a fraction of predicted, linear sum – not normalised by the standard deviation – and take the median of the distribution from each field of view. As we describe in the revised Figure 4Figure , this measure is more robust to noise. Notably, our finding that mixture summation is generally less sublinear in awake mice still stands for the early phase.

In the revised manuscript, we use the median fractional deviation whenever we compare linearity of summation across different conditions, which includes the comparison of anaesthetised vs. awake, behaving conditions (revised Figure 4), comparison of dendrites vs. somata (revised Figure 4—figure supplement 1), and comparisons of awake states (revised Figure 6). This has given us, too, more confidence about our interpretation, so we are grateful for the reviewer’s suggestions.

2. Related to the above issue, it would be useful to analyze the difference between conditions using different metrics to fully understand what really are different between conditions. The scatter plots shown in various figures do not show drastic differences between awake and anesthetized conditions, as might be indicated by the percent of sublinear responses. It would be useful to characterize the magnitude of sublinear/supralinear effects. For example, one can calculate a fractional change in the mean response. Does this measure show consistent difference between awake and anesthetized conditions?

Thank you for suggesting this analysis. As described in Figure 4 , we now use the fractional deviation to quantify how mixture summations differ from linear sums, which turned out to be a very useful way to express the property of summation (N.B.: noise is amplified for small responses when fractional deviation is used, which is another reason we use the median now). We thank the reviewer for suggesting this analysis.

Reviewer #2:This study addresses how complex stimuli are represented in neural responses. This is particularly relevant to olfaction because the vast majority of stimuli are complex mixtures that perceptually, are not easy to decompose into parts. Nonetheless, the ability to discern a relevant odor from background odors is essential. This process is easier when neural responses to mixtures reflect the linear sum of the responses to the individual components. The main conclusion of this study is that the linearity of olfactory bulb responses to two-component mixtures increases awake versus anesthetized states. The authors provide some evidence to support this claim. However, this could be better quantified and there is a temporal aspect of linearization that is not addressed. Perhaps the most interesting aspect of the study is the difference in linearity between the dendrites and the somata of the mitral/tufted cells. But a statistical analysis of this finding was not evident. Overall, a mechanistic or functional approach to understanding these findings is lacking. The differences linearity between the anesthetized and awake are simply explained by response saturation anesthetized animals. There are hints at mechanism by which linearity is supported in the OB with comparisons between soma and dendrite, but these are not well developed. There is a model that addresses the functional significance of linearity, but this is only supplemental and not well described.

Thank you for appreciating the significance of our work, and for your constructive comments. Please see below for our point-by-point responses to your concerns.

1) The quantification of linearity is incomplete. The deviation from linearity is a fine metric but just reporting the change in fraction of ROIs that show deviation from linearity in the sublinear direction is insufficient. There are many ROIs that show supralinear summation as well. Moreover, it is not clear where the cut-off is between linear and non-linear responses. There are statistical analyses that can be done on proportional data to determine if the proportions of non-linear ROIs are statistically different in the anesthetized and awake states. These should be done.

Thank you for raising this point. The problem with the way we quantified our data previously was raised by other reviewers, too, and we agree. In the revised manuscript, first, we refrain from using arbitrary cut-offs. Using the median fractional deviation (please see responses to Distilled Point #1 far above for details), we plot cumulative histograms to compare groups, which avoids forcing arbitrary categories (i.e., sublinear, linear, and supralinear). Please see below for the comparison of cumulative histograms (Figure 4).

2) Another approach is to compare the slope of the lines fit to data in the Observed vs. Linear sum plots and to determine how much these deviate from 1, and if this is statistically significant. These slopes could also be compared between states.

Thank you for suggesting the alternative analysis. We hope that our new measure, the fractional deviation, addresses the concern.

3) It is also necessary to be clear about which timepoint is being analyzed. It is apparent from figure 4 that the linearity of the population of ROIs evolves with time, becoming most linear 3 sec after odor onset. This temporal evolution of linearity is not addressed at all. But could provide some insight into mechanism.

Thank you for your suggestion to clarify our presentation. Whenever a single time point is presented, we annotate in the figure which time point the analysis relates to (Figures 4E,F; Figure 5A, Figure 4—figure supplement 1 D).

4) There is no statistical analysis of the dendrite versus soma data. The figure is not very convincing of differences. Why wasn't the temporal progression of responses shown for the dendrite versus soma data?

We apologise for not describing the dendrite data as fully. We now include the time course information for dendrites and somata in a way that is easier to relate the amplitudes to the linearity of summation (Figure 4 – supplement 1). The difference between the dendrites and somata seems to relate to the time course of activation somewhat – in general, Ca^2+^ rises earlier for the apical dendrites, which seems to manifest in the earlier appearance of sublinearity in this structure. We now also include data from MC somata to further understand the dendritic signal.

5) Linearity seems very correlated with the strength of the response. Sublinearity increases with strength suggesting saturation is a major contributing factor. Conversely, supralinear responses are more apparent with weaker component responses. It is well known that OB responses are much stronger in anesthetized animals. The deltaF/F values are greater in the anesthetized state compared to the awake state. So, the differences between the states may be trivially explained by saturation (as shown in Supplemental Figure 6). This doesn't really say anything about how the OB linearizes responses. However, the temporal evolution of linearization could speak to other mechanisms such as adaptation, recruitment of inhibition, descending control etc. Analyses of the dendrite versus soma signals over time could give insight as to where/when these mechanisms may be occurring. I think this was attempted in the supplement to figure 7, but the figure does not have sufficient explanation to figure out what is going on.

Thank you for the valuable comments. We agree that a deeper mechanistic insight would be useful. Indeed, as the reviewer suggests, the time course can give useful hints. To address this, we include the time course of response amplitudes next to the linearity of summation in our new supplemental figure (Figure 4, supplement 1). There is a rough correspondence between these two factors – as the reviewer points out, in general, the larger the response, the more sublinear the summation. However, the correspondence is not perfect and suggests something else at play. We additionally considered if correlation between component responses (i.e., EB vs. MB response correlation) could explain how linear the summations may be (Figure 4, supplement 1 panel E). This analysis suggests that pattern similarity may contribute, but again, not completely. We suspect that a comprehensive model of olfactory bulb circuits, with perhaps piriform feedback, may be needed to fully understand how the olfactory bulb processes mixture information. While this is somewhat beyond the scope of the current study, we hope to address it in our future investigations. Importantly, however, we hope that the additional data and analyses help.

6) There was a model of how linearity might function in odor detection, but this is not well described or motivated in the text.

Thank you for this helpful comment. We were not sure which model the reviewer was referring to, but we now discuss a recent work by (Reddy et al., 2018) where antagonism can, in principle, bring benefits to odour detection tasks, in addition to discussing the difficulty in mixture analysis due to nonlinear interactions. Thus, our introduction is more balanced, but the fact remains that we need to understand how mixtures are represented in behaviourally relevant context. We hope that the reviewer finds our revised description better. The revised introductory text is as follows (lines 68-70): “Nonlinear summation poses a difficulty because it may distort a pattern of interest brought by non-uniform addition of unpredictable background patterns, although more recent studies suggest beneficial effects of antagonism, for example by reducing saturation-related loss of information (Reddy et al., 2018).”, followed by (lines 75 – 76): “The question therefore remains: how does the mammalian olfactory system deal with nonlinear summation of responses?”

7) Overall, all the supplemental figures and data are essential to the study. Many of my concerns are somewhat addressed by these additional analyses. But they are not well explained. These should be integrated into the main figures of the paper and adequately described in the Results section.

Thank you very much for valuing our analyses that we had presented in the supplement. By addressing many excellent comments from the reviewers, our revised manuscript has been heavily edited. With the changes, we believe the original supplementary figures are either modified substantially, incorporated better, or are no longer needed.

Reviewer #3:Adefuin et al. use multiphoton imaging of M/T cell responses to investigate whether neuronal representations of binary mixtures can be explained as a sum of the components. The current view in the field (built largely from studies in anesthetized animals), is that mixture summation is non-linear and increases with the degree in glomerular response overlap elicited by the components. The authors reproduce these results and ask whether the same phenomenon is observed in the awake state, in particular when the animals are engaged in an odor discrimination task. Unlike in the anesthetized state, the authors find that mixture representations are linear in the awake brain. They use a series of systematic behavioral paradigms to show that the observed linearity in the awake state (compared to anesthetized) is not dependent on task engagement (reward is given randomly, post-odor) or stimulus relevance (reward is given before odor). While the experiments are well done and the data is presented clearly, I have several major concerns about the interpretation of their results.1) Given the data the authors present, it is unclear if one can conclude that the olfactory system is more or less linear in the awake state compared to the anaesthetised one. What seems to change most across the awake vs. anesthetized state is the response amplitude. Responses appear to be ~3x smaller in the awake mice. In the anesthetized state, non-linearity seems most apparent for large response amplitudes (>5 dF/F) with mixture responses being sub-linear, most likely due to saturation effects. The authors themselves do an analysis in Figure 6 – supplement 1 to show that most of the observed non-linearity in the anesthetized animals can be explained away after accounting for amplitude normalisation. The authors use this analysis to comment that the level of linearity is the same across all the three awake states, but the same figure shows that it is in fact the same even for the anaesthetized state.To put it differently, it is indeed true from the authors data that the OB response gain is significantly lower in the awake state, but it is unclear if the summation is more linear if measured at similar response amplitude regimes in both awake and anaesthetised mice.

Thank you for the valuable comments. We agree that many differences between the anaesthetised vs. awake states should have been taken into account when comparing the linearity of summation. We address the reviewer’s concern now by expressing the deviation as a fraction of the predicted, linear sum of component responses. Further, we also considered another factor that could influence the anaesthetised vs. awake comparison, namely, the trial-by-trial variability.

2) The authors argue that keeping response amplitudes small in the awake brain prevents sub-linear summation and therefore may lead to better mixture decomposition. They do a decoding analysis in anaesthetised mice to show that linear mixture representations (instead of using observed sub-linear representations) make odor classification easier. However, I find this analysis uninformative and misleading. It is no surprise that the decoders trained on single odor representations should perform better (or equivalent) when using linear sums as input instead of observed sub-linear representations. The authors use this observation to suggest that this mechanism aids discrimination ability in the awake state. However, given that even the single odor responses are much weaker and noisier in the awake state, it is likely that even the single odor discrimination ability is poorer in the awake state. By the same logic, mixture decomposition might be also much poorer in the awake brain than the anesthetized brain, even though summation is more linear, just because responses are weaker and noisier. In my opinion, the authors should compare decoding accuracy across awake vs. anesthetized responses if they want to assert that linearisation of responses in the awake brain leads to easier decomposition. Because otherwise, while linearisation in principle can aid decomposition, at least in the form that the authors observe here, it may come at a high cost on signal-to-noise ratio which would undo the gain that linearity provides, in principle, for discrimination.

Thank you very much for the insight and for the excellent suggestion to consider the discriminability of stimuli. In particular, we now include an analysis where a decoder trained on single responses is tested on observed mixture responses. Surprisingly, despite the substantial differences in the amplitudes of response and trial-by-trial variability, decoders using data from awake mice performed well, even better than anaesthetised data for the late phase of responses. This is now described in the revised figures (revised Figure 5). We thank the reviewer for the excellent suggestion.

Interestingly, though, the time course of the decoder performance does not correlate well with the linearity of summation. This observation is now described in the abstract (lines 19-21): “…decoding analyses indicated that the data from behaving mice was able to encode mixture responses well, though the time course of decoding accuracy did not correlate with the linearity of summation“.

3) At a more philosophical level, to this Reviewer, it is unclear if anesthesia vs. awake state difference in response should constitute the main focus of the manuscript. The authors explore summation properties under four different brain states, one of which is anaesthesia (also least behaviorally relevant). In three out of four states, they observe that summation is linear. In the fourth (anaesthesia), they observe that summation is sub-linear, but this happens at much larger response amplitude regimes compared to the three awake states sampled, presumably due to saturation. To me, it seems that the Authors here show that mixture summation in the OB, is largely independent of brain state since it is unaffected by whether the animal is task engaged or motivated etc.

Thank you for this thoughtful comment. This has made us reflect on the essence of our study. We believe we make three main observations. First, the anaesthesia vs. awake difference in the property of summation differ, and should be reported, because of the large volume of prior works reporting sublinear summations. However, as the reviewer recommends and as mentioned next, this is no longer the sole focus of our study. Our second observation is that the linearity of summation does not necessarily correlate with the ability to analyse mixtures, based on the decoder performance. We believe it is important to share this observation, since a number of previous studies speculated that nonlinear summation contributes to perceptual difficulty (Bell et al., 1987; Laing, 1994). Third, the decoder performance – especially one that is trained on single odour responses and tested on mixtures – shows differences depending on the awake states, where data from disengaged mice performed particularly poorly. This result is shown in the revised Figure 6Figure . Further, we have edited the abstract and results to ensure that these are clearly communicated. We hope that this is more balanced and reflects the data better.

4) It is unclear how to interpret the dendritic imaging comparison. First, the dendritic signal is pooled across many cells. If any of the cells that are being pooled shows sub-linearity, the pooled population response will look sub-linear, albeit less so than at the single cell level. Second, again like for the anesthetized vs. awake comparison, there is a discrepancy in response amplitudes – dendritic responses are ~2x stronger than the somatic responses and sub-linear summation would be more apparent as one approaches the saturation regime. Third, dendritic responses pool both mitral and tufted, while the somatic data the authors present is predominantly from tufted cells.

[Editors' note: further revisions were suggested prior to acceptance, as described below.]

Although the manuscript has greatly improved, there are some remaining issues. Specifically, Reviewer 2 and 3 have raised concerns that need to be clarified. In particular, Reviewer 3 has raised the issue regarding the dissociation between discriminability and linear mixture summation. The functional significance of linear mixture summation, or the limitation thereof, needs to be discussed more carefully, and it would be useful to include some discussion on potential causes of increased discriminability in awake conditions. In addition, Reviewer 2, raised several points about clarifying and adjusting statements in the abstract and the text of the manuscript, and analyzing the temporal dependence of decoding accuracy to the instantaneous level of non-linearity in response when comparing across different brain states. We would like you to respond to these remaining issues.

Thank you for recognising the significant improvement with the previous revision.

In this revision, we addressed the remaining concerns by providing additional analyses and text changes. Key to this was to follow Reviewer 1's helpful suggestion, namely, to model the functional consequence of nonlinear summation. Specifically, we simulated the normalising sublinearity on simulated linear sums of mixture responses and assessed how the SVM performance is affected by this. This provided a missing link that now allows us to interpret the discriminability of mixture responses and isolate the functional consequence of non-linear summation more clearly.

Generally, our simulation indicates that the sublinear mixture summation due to saturating influence limits the discriminability of mixture responses. Thus, dampened responses in awake animals bring some advantage by maintaining the mixture responses in the linear range, especially in the early phase. However, when evoked response patterns are decorrelated in the late phase, the saturating sublinearity is less detrimental to discrimination.

The simulation, as well as a graphical summary of our findings, are shown in a new figure (Figure 7).

Further, to explain our findings, particularly in light of our new analyses, we edited the abstract, relevant sections of the result and discussion are also revised concordantly as detailed below.

Lines 17- 27 (abstract): ".… we found that mixture summation is more linear in the early phase of evoked responses in awake, head-fixed mice performing an odour detection task, due to dampened responses. Despite this, and responses being more variable, decoding analyses indicated that the data from behaving mice was well discriminable. Curiously, the time course of decoding accuracy did not correlate strictly with the linearity of summation. Further, a comparison with naïve mice indicated that learning to accurately perform the mixture detection task is not accompanied by more linear mixture summation. Finally, using a simulation, we demonstrate that, while saturating sublinearity tends to degrade the discriminability, the extent of the impairment may depend on other factors, including pattern decorrelation. Altogether, our results demonstrate that the mixture representation in the primary olfactory area is state-dependent, but the analytical perception may not strictly correlate with linearity in summation."

Lines 240- 256 (results): Under what circumstance does discriminability of odour mixtures decouple from linearity in mixture summation? To study the effect of saturating sublinearity in isolation, we made a simulation as follows. Linear sums of responses were constructed by adding component responses obtained from the imaging sessions. In one test, these sums, with added noise, were passed through SVMs that had been trained with component responses. In another test, the linearly summed responses with noise were transformed with a normalising function (Mathis et al., 2016; Penker et al., 2020), which adds sublinearity in an amplitude-dependent manner (Figure 7B; see methods). Then, these signals were passed through the same SVMs to assess how discriminable the activity patterns were. When the data from the anaesthetised mice was used, normalising sublinearity was particularly detrimental around 2 seconds after the odour onset (Figure 7C), qualitatively reproducing the transient decrease in the accuracy seen with the observed mixture responses (Figure 5E). In contrast, with the data from the behaving mice, normalising sublinearity did not have as significant an effect on the mixture discriminability, even at the later stage when sublinear summation becomes more widespread (Figure 7D,E). Thus, the functional consequence of nonlinear summation may depend on specific circumstances, for example on how separable, or decorrelated, the activity patterns are. Overall, our result indicates that mixture responses in the olfactory bulb are highly state-dependent and evolve over time (Figure 6F)".

Lines 272-285 (discussion): " This raises a question: is sublinear summation in mixtures detrimental, or beneficial to discriminating mixtures of odours? Our simple simulation indicates that, generally, a sublinear mixture summation due to saturating influences limits the discriminability of mixture responses. Thus, dampened responses in awake animals bring some advantage by maintaining the mixture responses in the linear range. However, in the behaving mice, even when the responses became larger at a later phase, the saturating influence was less detrimental to discrimination. Here, decorrelated response patterns may play a more crucial role, as it is known to enhance classifier performances (Bhattacharjee et al., 2019; Friedrich and Wiechert, 2014; Gschwend et al., 2015; Padmanabhan and Urban, 2010). While the behavioural task described in this study is simple and animals make accurate decisions within 1 s (771 ± 97 ms) of stimulus onset, the slower mechanisms described here may be important when larger and more reliable responses are required for accurate decisions. So, in addition to sampling time (Rinberg et al., 2006), this phenomenon may be a part of mechanisms needed to solve more difficult tasks accurately (Abraham et al., 2010; Wilson et al., 2017). "

We hope that these additional analysis, simulation, and text changes address the remaining concerns and that the findings are explained more clearly.

Reviewer #1:The authors have addressed my previous concerns by reanalyzing the data and revising the manuscript. Importantly, the authors now provide a more convincing metric for linearity of mixture interaction using the median fractional change over the population. I am more convinced by this metric which supports that mixture interaction is more linear in awake conditions. This new analysis addresses my most important concern. A new analysis also finds that decoding or behavioral ability to detect a mixture component did not correlate with the linearity of mixture interactions in odor responses. This finding, in a sense, makes the significance of the finding of linearity in odor responses a little unclear. On the other hand, I think that the demonstration that odor decoding does not benefit from linear mixture interaction has an important message to the field as one might assume that it would be the case. It would be more useful if the authors can demonstrate this point (odor decoding does not benefit from linear mixture interaction) in a more abstract form without relying on the data -- can we take this conclusion as a general result, or does it rely on some particular features in the data. A simple modeling might help address this issue. Although such demonstration is not required, if the authors can address this in a short period of time, it would enhance the manuscript significantly, although this point is a little secondary compared to the main point (linearity of odor mixture interaction).Overall, I am satisfied by the more convincing demonstration of linearity of mixture interaction. Because sublinearity has been emphasized in the previous literature, the results presented in this manuscript provide an important result.

Thank you for noting that our manuscript had improved, and for your valuable inputs. In particular, the suggestion to model the nonlinearity to understand the functional consequence (on discriminability) proved to be key to interpreting the result. Generally, our simulation indicates that the sublinear mixture summation due to saturating influence limits the discriminability of mixture responses, but the extent of impairment may depend on other factors such as decorrelation.

Reviewer #2:I commend the authors for re-structuring the manuscript and performing additional analyses to address the concerns raised. These changes have indeed improved the manuscript substantially.However, I still have concerns about the disparity in response amplitude and dynamics across the different states and how that may affect the interpretation of the results. I think these can be addressed by further clarifying and editing the statements in the abstract and rest of the manuscript before publication.1. The authors use the new metric – Fractional deviation – to compare differences in the degree of linearity between the anesthesia and awake states. While this metric normalizes deviations from linearity with the response amplitude, it does not correct for the overall difference between the means of the two response amplitude distributions – in anesthesia versus awake states. If majority of the responses during the awake state are smaller than during anesthesia, and the amount of non-linearity scales with response amplitude – irrespective of the metric chosen, there will be overall more non-linear instances in anesthesia than in the awake state. This does not imply that large amplitude responses will sum 'more' linearly in the awake state. To this reviewer, an adequate comparison to test this possibility is sub-sampling the response distribution under anesthesia and comparing differences in non-linearity between responses of similar amplitudes. Do similar amplitude responses tend to sum more, or less linearly in awake versus anaesthetized states?I think that the observation that, in response to the same stimuli, at the level of mitral and tufted cells, are kept in the linear regime of the system is insightful and would be important to document. It would enable the field to move forward. In my opinion, one way for the authors to clarify this matter is to show both the normalized and un-normalized data in the manuscript and simply state that in the awake state, responses to the same stimuli tend to be smaller in amplitude than under anesthesia, and, therefore, remain in the linear summation regime (i.e. do not get to reach high amplitudes that would sum up in a more non-linear fashion). This could be due to, for example, to stronger gain control mechanisms in the awake state, etc.

Thank you for your valuable comments. We apologise that our explanations were not clear enough. Essentially, we believe that dampened responses make the mixture responses in the awake state to remain in the linear range, which, if we understood correctly, agrees with your intuition. We now explain this as a graphical summary (Figure 7E). We hope that this presentation is clearer. Thank you for raising this point.

2. The authors added analysis to compare the discriminability between the anesthesia and awake state. While it is reassuring that discriminability is decent even with the smaller amplitude responses in the awake state. I find that the overall interpretation of the results could potentially lead to confusion in the readers' minds. Therefore, I propose to clarify by adding a few explanatory statements, and adjusting some of the statements in the abstract.If I understand correctly, the authors use a 0.5s pulse of odor and find that discriminability during the first second or so, is comparable (for training an testing on random selection of single odor responses and mixture responses) or better (for training on single odors, testing on mixtures) during anesthesia. In the later phase (>1.5 s), awake responses appear to do better at discriminability. I am not sure how to interpret this finding. Could this simply be related to differences in response amplitudes and temporal dynamics? If so, stating this, would be very useful to bring clarity to the message. From the examples in Figures 1,2,4 it appears that responses during anesthesia peak early and decay back to baseline within 1 second from odor OFF (so, mostly back to baseline at 1.5 s from odor ON), whereas responses during the awake state have longer latencies and peak later. Therefore, the time-course differences in responses might explain the time-course difference in discriminability. It would be important to state that these differences in response dynamics across the two states may be the basis for the time differences in discriminability.

Thank you for highlighting the potential source of confusion. To address this point, we have edited the abstract in the following manner.

Lines 16-21: " Unlike previous studies using anaesthetised animals, we found that mixture summation is more linear in the early phase of evoked responses in awake, head-fixed mice performing an odour detection task, due to dampened responses. Despite this, and responses being more variable, decoding analyses indicated that the data from behaving mice was well discriminable. Curiously, the time course of decoding accuracy did not correlate strictly with the linearity of summation.".

In addition, we added the following text in the discussion as below.

Lines 275-280: " Thus, dampened responses in awake animals bring some advantage by maintaining the mixture responses in the linear range. However, in the behaving mice, even when the responses became larger at a later phase, the saturating influence was less detrimental to discrimination. Here, decorrelated response patterns may play a more crucial role, as it is known to enhance classifier or behavioural performances (Bhattacharjee et al., 2019; Friedrich and Wiechert, 2014; Gschwend et al., 2015; Padmanabhan and Urban, 2010).".

We hope that the temporal evolution of the responses, their effects on the properties of mixture summation, as well as the functional consequence, are now better explained.

From a behavioral point of view, it is unclear how to interpret these time-courses, discriminability peaks much later than behaviorally relevant time-scales in the awake state (<1.5s for the odor concentration used, 1-5%), and while odor identities can be decoded in anesthesia – animals don't make decisions under KX. When comparing the discriminability between naïve engaged, it would be important to clarify whether these differences arise early, at behaviorally relevant time-scales, or only appear in the later phase of the responses (>1.5 s) from odor onset and adjust the statements in the abstract accordingly. A side by side comparison would clarify this point.

Thank you for this comment. While, indeed, the discriminability peaks later, the SVMs start to perform above chance at an earlier time point. It is well possible that this is enough for the downstream areas to lead to good behavioural performance level. It should be noted that, in the imaging sessions, we sample from a very limited number of the output neurons, whereas the brain has access to all OB outputs. That is, the discriminability could reach a good level even earlier if more informative neurons were included. To clarify this timing point, we added the following text in the result section:

Lines 192-195: ".…M/T cells from the two conditions performed similarly for the first 1 second, performing above chance soon after the odour onset (earliest time where accuracy is significantly above 0.5 = 0.67 s for behaving case, and 1.12 s for anaesthetised data; t-test at significance level of 0.05; n = 13 fields of view, 6 mice for behaving case; 8 fields of view, 4 mice for the anaesthetised case). ".

3. Discriminability as a function of linearity of summation. It is unclear what the authors mean by 'the time course of decoding accuracy did not correlate with the linearity of summation'. The authors do not quantify linearity as a function of time, or explicitly analyze the temporal dependence of decoding accuracy with the instantaneous level of non-linearity. To support the above-mentioned sentence, such analysis would be necessary. Additionally, while this is a nice result (that decoding accuracy is not a direct correlate of linearity), it dampens the significance of assessing differences in linearity between various brain states.

Thank you for this suggestion. We have added a supplementary figure (Figure 6—figure supplement 2) to address this. Specifically, we plotted the instantaneous relationship between the SVM performance and the linearity of summation to direct comparison .

Reviewer #3:The authors have made a good attempt to address concerns raised in the first review. They have reanalyzed the data and used an alternate metric- mean fractional deviation, to evaluate response linearity. Importantly their original result still holds, M/T responses are more linear in the awake state than anesthetized. There is still some issue with stating the importance of this finding. It still reads like a non-finding from a functional perspective. Linearity only seems to be helpful with respect to discrimination in a time frame (>2s) that is after the animal has already started licking. But it is also important to note that despite a reduction in discrimination in early odor responses compared to anesthetized this does present a problem at the behavioral level. Particularly for a simple task such as the one used here. The longer time frame maybe important for more difficult tasks. Also, a potentially underestimated finding is the overall reduction in correlation between M/T cells in the awake state (Figure 5B). Such a reduction in correlation decreases redundancy which may enhance coding (Padmanbhan and Urban, 2010). These points should be emphasized more.

Thank you for noting the improvement. Regarding the main message, we agree that it needs clarification. We were able to clarify, with simulation, that saturating sublinearity is generally detrimental, but at a later phase, when sublinear summation occurs in conjunction with other phenomena like decorrelation, it may not affect the discriminability as much. Due to the larger responses at late time points, the benefit of linear summation seems to be most relevant in the early phase (t < 1.5s), which is when patterns are less decorrelated. This is now described directly in the new Figure 7.

We are grateful for your acute observation that pattern decorrelation may be a contributing factor. We have edited the abstract and the discussion to reflect this, adding the reference you cited above. In addition, we have also edited the discussion to incorporate the potential relevance of late onset mechanisms (i.e., decorrelation and peak in the discriminability).

Text additions/changes:

Lines 21- 25 (abstract): "Finally, using a simulation, we demonstrate that, while saturating sublinearity tends to degrade the discriminability, the extent of the impairment may depend on other factors, including pattern decorrelation."

Lines 240- 156 (results): Under what circumstance does discriminability of odour mixtures decouple from linearity in mixture summation? To study the effect of saturating sublinearity in isolation, we made a simulation as follows. Linear sums of responses were constructed by adding component responses obtained from the imaging sessions. In one test, these sums, with added noise, were passed through SVMs that had been trained with component responses. In another test, the linearly summed responses with noise were transformed with a normalising function (Mathis et al., 2016; Penker et al., 2020), which adds sublinearity in an amplitude-dependent manner (Figure 7B; see methods). Then, these signals were passed through the same SVMs to assess how discriminable the activity patterns were. When the data from the anaesthetised mice was used, normalising sublinearity was particularly detrimental around 2 seconds after the odour onset (Figure 7C), qualitatively reproducing the transient decrease in the accuracy seen with the observed mixture responses (Figure 5E). In contrast, with the data from the behaving mice, normalising sublinearity did not have as significant an effect on the mixture discriminability, even at the later stage when sublinear summation becomes more widespread (Figure 7D,E). Thus, the functional consequence of nonlinear summation may depend on specific circumstances, for example on how separable, or decorrelated, the activity patterns are. Overall, our result indicates that mixture responses in the olfactory bulb are highly state-dependent and evolve over time (Figure 6F). ".

Lines 272-285 (discussion): " This raises a question: is sublinear summation in mixtures detrimental, or beneficial? Our simple simulation indicates that, generally, a sublinear mixture summation due to normalising influences limits the discriminability of mixture responses. Thus, dampened responses in awake animals bring some advantage by maintaining the mixture responses in the linear range. However, in the behaving mice, even when the responses became larger at a later phase, the normalising effect was less detrimental to discrimination. Here, decorrelated response patterns may play a more crucial role, as it is known to enhance classifier performances (Friedrich and Wiechert, 2014; Gschwend et al., 2015; Padmanabhan and Urban, 2010). While the behavioural task described in this study is simple and animals make accurate decisions within 1 s (771 ± 97 ms) of stimulus onset, the slower mechanisms described here may be important when larger and more reliable responses are required for accurate decisions. So, in addition to sampling time (Rinberg et al., 2006), this phenomenon may be a part of mechanisms needed to solve more difficult tasks accurately (Abraham et al., 2010; Wilson et al., 2017). "

We hope that the main message is now clear.